# The effect of particle acidity on secondary organic aerosol formation from α-pinene photooxidation under atmospherically relevant conditions

Yuemei Han, Craig A. Stroud, John Liggio, and Shao-Meng Li

Air Quality Research Division, Atmospheric Science and Technology Directorate, Environment and Climate Change Canada, Toronto, ON, M3H 5T4, Canada

*Correspondence to*: Craig A. Stroud (craig.stroud@canada.ca) and Yuemei Han (yuemeihan@hotmail.com)

**Abstract.** Secondary organic aerosol (SOA) formation from photooxidation of α-pinene has been investigated in a photochemical reaction chamber under varied inorganic seed particle acidity levels at moderate relative humidity. The effect of particle acidity on SOA yield and chemical composition was examined under high- and low-$NO_x$ conditions. The SOA yield (4.2%−7.6%) increased nearly linearly with the increase in particle acidity under high-$NO_x$ conditions. In contrast, the SOA yield (28.6%−36.3%) was substantially higher under low-$NO_x$ conditions, but its dependency on particle acidity was insignificant. A relatively strong increase in SOA yield (up to 220%) was observed in the first hour of α-pinene photooxidation under high-$NO_x$ conditions, suggesting that SOA formation was more effective for early α-pinene oxidation products in the presence of fresh acidic particles. The SOA yield decreased gradually with the increase in organic mass in the initial stage (approximately 0–1 hour) under high-$NO_x$ conditions, which is likely due to the inaccessibility to the acidity over time with the coating of α-pinene SOA, assuming a slow particle-phase diffusion of organic molecules into the inorganic seeds. The formation of later-generation SOA was enhanced by particle acidity even under low-$NO_x$ conditions when introducing acidic seed particles after α-pinene photooxidation, suggesting a different acidity effect exists for α-pinene SOA derived from later oxidation stages. This effect could be important in the atmosphere under conditions where α-pinene oxidation products in the gas-phase originating in forested areas (with low $NO_x$ and $SO_x$) are transported to regions abundant in acidic aerosols such as power plant plumes or urban regions. The fraction of oxygen-containing organic fragments ($C_xH_yO_1^+$ 33–35% and $C_xH_yO_2^+$ 16–17%) in the total organics and the O/C ratio (0.52–0.56) of α-pinene SOA were lower under high-$NO_x$ conditions than those under low-$NO_x$ conditions (39–40%, 17–19%, and 0.61–0.64), suggesting that α-pinene SOA was less oxygenated in the studied high-$NO_x$ conditions. The fraction of nitrogen-containing organic fragments ($C_xH_yN_z^+$ and $C_xH_yO_zN_p^+$) in the total organics was enhanced with the increases in particle acidity under high-$NO_x$ conditions, indicating that organic nitrates may be formed heterogeneously through a mechanism catalyzed by particle acidity or that acidic conditions facilitate the partitioning of gas phase organic nitrates into particle phase. The results of this study suggest that inorganic acidity has a significant role to play in determining various organic aerosol chemical properties such as mass yields, oxidation state, and organic nitrate content. The acidity effect being further dependent on the time scale of SOA formation is also an important parameter in the modeling of SOA.

## 1 Introduction

Secondary organic aerosols (SOA) formed by oxidation of biogenic and anthropogenic volatile organic compounds (VOC) comprise a substantial portion of submicron aerosol particles in the atmosphere (Kanakidou et al., 2005; Zhang et al., 2007a). Understanding the physical and chemical properties associated with SOA formation and transformation is important to adequately assess aerosol impacts on climate and human health (Hallquist et al., 2009). The effect of aerosol acidity on SOA formation is one of the scientific questions currently under open debate, as described below. Acid-catalyzed heterogeneous reactions such as hydration, hemiacetal/acetal formation, polymerization, and aldol condensation have been proposed to form SOA (Jang et al., 2002). The presence of acidic aerosol particles has been reported to enhance the reactive uptake of gas phase organic species and increase SOA yields due to acid-catalyzed reactions (e.g., Garland et al., 2006; Jang et al., 2004; Liggio and Li, 2006; Lin et al., 2012; Northcross and Jang, 2007; Surratt et al., 2010; Xu et al., 2015a). However, other studies have suggested that those reactions may be thermodynamically or kinetically unfavorable and are possibly insignificant in the real atmosphere (Barsanti and Pankow, 2004; Casale et al., 2007; Kroll et al., 2005; Li et al., 2008). In contrast, recent kinetics studies have demonstrated that particle acidity strongly affects the reactive uptake of isoprene epoxydiols (Gaston et al., 2014; Riedel et al., 2015). Furthermore, the enhanced formation of SOA and organic sulfates has been reported from the acid-catalyzed reactive uptake of VOC oxidation products in ambient aerosols that are acidic enough to promote this multiphase chemistry (Hawkins et al., 2010; Lin et al., 2012; Rengarajan et al., 2011; Zhang et al., 2012; Zhou et al., 2012), which is contrary to other field studies showing no apparent evidence of acid-catalyzed SOA formation (Peltier et al., 2007; Takahama et al., 2006; Tanner et al., 2009; Zhang et al., 2007b). The dependence of SOA formation on aerosol acidity generally has not been incorporated in many atmospheric chemistry models thus far due to the large uncertainties associated with the quantification of acidity effects, with the exception of the acidity effect for SOA via isoprene epoxydiol uptake (Marais et al., 2016; Pye et al., 2013).

A number of laboratory studies have investigated the effect of particle acidity on SOA formation from oxidation of various precursor hydrocarbons such as isoprene, terpenes, toluene, m-xylene, and 1, 3-butadiene (e.g., Kristensen et al., 2014; Lewandowski et al., 2015; Ng et al., 2007a; Offenberg et al., 2009; Song et al., 2013; Surratt et al., 2007b). α-Pinene is the most abundant biogenic monoterpene emitted from terrestrial vegetation (Guenther et al., 2012). The oxidation of α-pinene by hydroxyl radicals (OH), ozone ($O_3$), and nitrate radicals produces a variety of multifunctional organic compounds such as carboxylic acids, carbonyls, peroxides, ester dimers, epoxides, alcohols, and organic nitrates (Calogirou et al., 1999; Yasmeen et al., 2012; Zhang et al., 2015). Despite the efforts of previous laboratory studies under various experimental conditions, the effect of aerosol acidity on SOA formation from individual hydrocarbons remains unclear due to the complexity of this scientific question. In particular, the magnitude of the acidic effect on SOA yields for α-pinene has been found to vary significantly. For instance, a nearly 40% increase in organic carbon (OC) was observed for the ozonolysis of α-pinene in the presence of acidic seed particles without $NO_x$, and aerosol acidity played an important role in the formation of high molecular weight organic molecules in particles (Iinuma et al., 2004). A linear increase of 0.04% in OC mass per

65  nmol $H^+$ $m^{-3}$ was reported from the photooxidation of α-pinene with $NO_x$, and this effect was independent of initial hydrocarbon concentration or the generated organic mass (Offenberg et al., 2009). In contrast, Eddingsaas et al. (2012) reported a relatively small increase of SOA yield (approximately 22%) for OH photooxidation of α-pinene under high-$NO_x$ conditions, and no effect of aerosol acidity on SOA yield under low-$NO_x$ conditions when introducing acidic seeds before photooxidation. Kristensen et al. (2014) similarly found that the increase of aerosol acidity has a negligible effect on SOA

formation from ozonolysis of α-pinene under low-$NO_x$ conditions.

These inconsistent results reported previously are most likely attributed to the varied experimental parameters such as particle acidity, initial hydrocarbon concentration, oxidant type and level, $NO_x$ level, temperature, and relative humidity (RH). Most previous studies were conducted with different acidity levels, and therefore a quantitative comparison of the acidity effect among various studies is difficult. Laboratory studies were usually performed with relatively high loadings of

hydrocarbons (e.g., from tens of ppb to several ppm), which would result in higher yield and lower oxidation state of laboratory SOA compared to ambient SOA (Ng et al., 2010; Odum Jay et al., 1996; Pfaffenberger et al., 2013; Shilling et al., 2009). The oxidant used in laboratory studies is also possibly one of the important factors affecting SOA formation. For example, a positive dependence of SOA yield on $H_2O_2$ level has been reported for the photooxidation of isoprene (Liu et al., 2016). In addition, the presence of $NO_x$ during α-pinene oxidation may change the reaction chemistry and lead to the

formation of relatively volatile oxidation products, and hence decrease α-pinene SOA yields (Eddingsaas et al., 2012; Ng et al., 2007a). Moreover, temperature is an important factor in SOA formation; higher SOA yields may be obtained at lower temperature (Saathoff et al., 2009; Takekawa et al., 2003). RH is another important factor, the decrease of which may lead to an increase in α-pinene SOA yields (Jonsson et al., 2006); however, many previous studies have been performed at very low RH (e.g., less than 10%) or even dry conditions. As a result of the above issues, it is highly important for laboratory studies

to investigate the acidity effect on SOA formation under more realistic conditions approaching those of the ambient atmosphere. This would facilitate an accurate parameterization of the acidity effect for incorporation into air quality models.

This study aims to improve our current understanding of the effect of particle acidity on SOA formation from photooxidation of α-pinene. Photochemical chamber experiments were performed under conditions with relatively low α-pinene loadings and moderate RH, which are more representative of the ambient atmosphere. The yield of α-pinene SOA

was obtained at various particle acidity levels under high- and low-$NO_x$ conditions. The dependence of SOA yield on particle acidity and the time scale of the acidity effect are characterized and discussed. The effect of particle acidity on the chemical composition of α-pinene SOA, the fragment distributions of bulk organics, and the oxidation state of organics are examined based on the high-resolution analysis of organic aerosol mass spectra. The possible contribution of particle acidity to the formation of particulate organic nitrates under high-$NO_x$ conditions is also discussed. Finally, the potential significance of

the observed acidity effect in the ambient atmosphere is summarized.

## 2 Experimental methods

Photooxidation experiments were performed in a 2 m$^3$ Teflon chamber (Whelch Flurocarbon) enclosed in an aluminum support (Liggio and Li, 2006; Liggio et al., 2005). Twelve black light lamps (model F32T8/350BL, Sylvania) were used as the irradiation source with intensity peaking at approximately 350 nm. The chamber was flushed by zero air with the lamps turned on for more than 20 hours before each experiment to avoid contamination from previous experiments. Hydrogen peroxide ($H_2O_2$) vapor was introduced into the chamber to produce OH radicals during the first 6 hours of flushing. Temperature and RH inside the chamber were monitored continually using a temperature and humidity probe (model HMP 60, Vaisala). Temperature was not controlled during the experiments, but it was relatively constant at 25 °C before experiments began and increased to a stable value (approximately 30–34 °C) after the lamps were turned on. RH was maintained manually by adding water vapors generated from a bubbler with zero air as a carrier gas (15 L min$^{-1}$). Other chamber inputs (e.g., $H_2O_2$ vapor, NO, seed particles, and α-pinene) were conducted after the RH reached approximately 60%. RH inside the chamber stabilized at approximately 29–43% after the lamps were turned on for about 1 hour due to the increase in temperature.

$H_2O_2$ vapor, as the source of OH radical, was introduced into chamber using a bubbler with a flow of zero air (0.09 L min$^{-1}$) passing through $H_2O_2$ aqueous solution (30 weight % in water, Sigma-Aldrich) for 1 hour. Ammonium sulfate (AS)/sulfuric acid (SA) solutions with varied $NH_4/SO_4$ molar ratios were used to provide various acidities in seed particles. A complete list of the composition of the seed particles and other initial conditions in all experiments is given in Table 1. The seed particles were generated by atomizing AS/SA aqueous solution using an aerosol generator (model 3706, TSI), dried in a silica gel diffusion dryer, and then size-selected at 150 nm in mobility diameter using a differential mobility analyzer (DMA, model 3081, TSI). Nitric oxide (NO) was added into the chamber from a compressed gas cylinder (9.1 ppm NO in nitrogen) in high-$NO_x$ experiments, in contrast to low-$NO_x$ experiments where NO was not added. A micro-syringe was used to inject approximately 0.25 μL liquid α-pinene (99+%, Sigma-Aldrich) into the chamber through a stainless steel tube with a zero air at 3 L min$^{-1}$. After achieving the desired experimental conditions for a stable 30 min period, photooxidation reactions were initiated by turning on the lamps. The typical photooxidation time was 6 and 15 hours for high- and low-$NO_x$ experiments, respectively.

Four experiments were also performed to investigate the effect of aerosol acidity on α-pinene oxidation products at different photooxidation stages (Exp. 9–12; Table 1). Photooxidation of α-pinene was conducted without seed particles in the reaction chamber for 2 and 4 hours under high- and low-$NO_x$ conditions, respectively. This was followed by turning off the lamps and adding neutral/acidic seed particles into the chamber within 1 hour. The experiments continued for another 6 hours on the reactive uptake of the α-pinene oxidation products by the newly introduced seed particles in the dark.

The concentration of α-pinene in the chamber was measured in real-time using a proton-transfer-reaction time-of-flight mass spectrometer (PTR-ToF-MS, Ionicon Analytik GmbH) (Hansel et al., 1999; Lindinger and Jordan, 1998). The mixing ratios of NO and $O_3$ were monitored using a NO analyzer (model 42i-Y, Thermo Scientific) and an $O_3$ monitor (model 202,

2B Technologies), respectively. The particle number size distribution was measured using a scanning mobility particle sizer (SMPS) consisting of a DMA (model 3081, TSI) and a condensation particle counter (model 3776, TSI). The non-refractory chemical composition of the submicron aerosol particles, including organics, sulfate, ammonium, nitrate, and chloride, was measured using a high resolution time-of-flight aerosol mass spectrometer (HR-ToF-AMS, Aerodyne Research) (DeCarlo et al., 2006). The AMS instrument was operated in a high-sensitivity mode (V-mode) with the data stored at 1 min intervals.

The AMS data were processed using the standard ToF-AMS data analysis software (SQUIRREL v1.56D and PIKA v1.15D, http://cires.colorado.edu/jimenez-group/ToFAMSResources/ToFSoftware/). The mass concentrations of aerosol species were generated from the PIKA analysis of raw mass spectral data. A collection efficiency value of 0.7 was applied for the AMS data analysis based upon the comparison of the volume concentrations derived from AMS and SMPS measurements, assuming that particles are spherical and the densities of organics, sulfate, ammonium, and nitrate are 1.4, 1.77, 1.77, and

1.725 g cm$^{-3}$, respectively. Note that aerosol particles were not dried upstream of the AMS and SMPS measurements, and thus particle water content might have contributed to the SMPS-derived volume concentrations. This was not taken into account for the AMS-derived volume concentration. The detection limits of organics, sulfate, nitrate, and ammonium, defined as 3 times the standard deviations of the mass concentrations of individual species (1-min average) in particle-free air, were 34, 4, 1, and 5 ng m$^{-3}$, respectively.

SOA yield, which represents the aerosol formation potential of precursor hydrocarbon, was calculated from the ratio of generated SOA mass ($\Delta M_0$) to the reacted α-pinene mass ($\Delta HC$). The SOA yields and $\Delta M_0$ presented in Table 1 correspond to the maximum values at the end of each experiment. Organic mass concentrations derived from AMS measurement were wall-loss corrected according to the decay of sulfate particles in the chamber, i.e., by multiplying the ratios of the initial sulfate concentrations to the instantaneously measured sulfate concentrations. This correction assumed that α-pinene

oxidation products condensed on the sulfate particles instead of their self-nucleation. This assumption is appropriate given that less than 50 particles cm$^{-3}$ were contributed by self-nucleation and that an obvious increase in organic mass concentration was not observed from the AMS measurement in the experiments without adding seed particles. The decay rate of particles coated with organics was assumed to be same as that of pure sulfate particles, although the later could be slightly higher due to the larger Brownian diffusion rate of smaller particles. The calculated SOA yield could have been

affected by the wall loss of vapors at low α-pinene loadings, in particular for low- and semi-volatile gaseous species (Ehn et al., 2014); however, such an effect was not taken into account herein.

The initial seed composition in each experiment was predicted using the Extended Aerosol Inorganic Thermodynamic Model (E-AIM) II (http://www.aim.env.uea.ac.uk/aim/aim.php) (Clegg et al., 1998). The concentrations of inorganic sulfate, nitrate, and ammonium derived from the AMS measurement as well as the temperature and RH in the chamber were input

parameters. The pH of aerosol particles was calculated by $-\log(\gamma \times [H^+])$ using the model outputs, where $\gamma$ and $[H^+]$ are the activity coefficient of $H^+$ and the molar concentration of dissociated $H^+$ (mol L$^{-1}$) in the aqueous phase, respectively. OH concentration in each experiment was estimated from a linear fitting of the first order decay of gaseous α-pinene by OH radicals, i.e., the difference between the total α-pinene decay and the α-pinene consumed by $O_3$, as described by Liu et al.

(2015). The OH concentrations were calculated to be approximately 4.3–5.9 × 10$^6$ and 0.8–1.1 × 10$^6$ molecules cm$^{-3}$ for experiments under high- and low-NO$_x$ conditions, respectively. Nitrate radical (NO$_3$) generated from the reactions such as NO$_2$ with O$_3$ might also affect the α-pinene decay (and hence the estimated OH), whereas it was not taken into account here because NO$_3$ levels were likely to be small under the studied irradiation conditions.

## 3 Results and discussion

### 3.1 α-Pinene SOA formation under high- and low-NO$_x$ conditions

An increase of α-pinene SOA mass concentration with the decay of α-pinene mixing ratio in high- and low-NO$_x$ experiments using (NH$_4$)$_2$SO$_4$ seed particles (Exp. 1 and 5 in Table 1) is shown in Figure 1. Under the high-NO$_x$ condition, the increase of α-pinene SOA mass was observed shortly after the irradiation started until the end of the experiment (Figure 1a). Gaseous α-pinene was mostly consumed within approximately 1.5 hours. NO (66 ppbv initially) was consumed in the first 30 min of the irradiation. The formation of O$_3$ was not suppressed over the experiment. O$_3$ increased to more than 200 ppb at the end of the experiments, and therefore ozonolysis reactions would have contributed to the formation of α-pinene SOA. A rough estimation shows that α-pinene consumed by ozonolysis accounted for in the range of 0–28% of the total α-pinene decay, as seen from the difference between the total α-pinene decay and OH consumed α-pinene in Figure 1a. Nitrate radicals may also have been generated from NO$_x$ reactions and have contributed to α-pinene SOA formation, whereas its direct measurement was not available in this study.

In contrast, under the low-NO$_x$ condition, the increase of SOA mass concentration and the decay of α-pinene were relatively slower (Figure 1b). This is most likely due to the lower production of OH radicals from H$_2$O$_2$ photolysis under low-NO$_x$ condition, that is, 1.1 × 10$^6$ molecules cm$^{-3}$ compared to that of 5.3 × 10$^6$ molecules cm$^{-3}$ under high-NO$_x$ conditions. A plateau of the generated SOA mass was observed after approximately 12 hours of irradiation, suggesting that SOA formation reached equilibrium after α-pinene was consumed completely, if the gas-particle partitioning was reversible (Grieshop et al., 2007). NO was less than 0.3 ppbv through the entire experiment. A slight increase of O$_3$ (up to 30 ppb) was also observed under low-NO$_x$ conditions, which might have resulted from the photolysis of a small amount of NO$_2$ released from the chamber walls. Less than approximately 47% of α-pinene was consumed by ozonolysis.

The α-pinene SOA yield was 4.2% when using (NH$_4$)$_2$SO$_4$ seed particles under high-NO$_x$ condition, which is a factor of 8.4 lower than that under low-NO$_x$ condition (35.2%) (Table 1). The relatively lower SOA yield under higher NO$_x$ levels is consistent with those reported previously for the photooxidation and ozonolysis of α-pinene (Eddingsaas et al., 2012; Ng et al., 2007a; Presto et al., 2005). Similar relationships are also observed in the photooxidation of isoprene and aromatic hydrocarbons such as benzene, toluene, and *m*-xylene (Kroll et al., 2006; Ng et al., 2007b). The dependence of α-pinene SOA yield on NO$_x$ level is possibly due to the different gas-phase chemical reactions of the intermediate organic peroxy radicals (RO$_2$) formed in the initial photooxidation stage. RO$_2$ reacted primarily with NO under high-NO$_x$ conditions and

generated relatively more volatile products that reduced the overall SOA yield, whereas the reactions of $RO_2$ with other peroxy radicals (e.g., $RO_2$ and $HO_2$) were dominant under low-$NO_x$ conditions (Kroll et al., 2006; Presto et al., 2005; Xu et al., 2014). Approximately 62–99% of $RO_2$ radicals reacted with NO over the entire experimental time (totally 6 hours) under high-$NO_x$ conditions in this study, which was estimated based on the Master Chemical Mechanism constrained by the initial experimental conditions (S1 and Figure S1 in the Supplement). The observed difference in SOA yields might also be affected to some extent by other experimental conditions such as the initial α-pinene concentration, seed loading, and temperature, but $NO_x$ level was most likely the primary cause, given that other factors did not vary as much as $NO_x$ in these experiments (Table 1).

A comparison of SOA yields as a function of the generated SOA mass ($\Delta M_0$) in this study and previous studies of α-pinene photooxidation is shown in Figure 2. The experimental parameters and SOA yields from previous studies are summarized in Table 2. SOA yields in these studies varied in the range of approximately 1.3–24% in the presence of $NO_x$, in contrast to those of 26–46% without $NO_x$. The SOA yields observed in our study are generally comparable to those reported from previous studies despite the different experimental parameters. The SOA yield under high-$NO_x$ condition in our study is in particular closest to those with lower α-pinene level (Ng et al., 2007a; Odum et al., 1996) and at higher temperature (Takekawa et al., 2003). It has been established that low α-pinene level and high temperature can lead to relatively low SOA yields, which is possibly due to the changes of gas/particle partitioning thermodynamics in the reaction system (Odum et al., 1996; Pankow, 2007; Takekawa et al., 2003). The SOA yield under low-$NO_x$ conditions here is similar to those reported by Eddingsaas et al. (2012) but lower than those reported by Ng et al. (2007a). The varied SOA yields among these studies most likely depend on different experimental conditions. Therefore, laboratory studies performed under conditions relevant to the atmosphere are important for an inter-comparison among studies and for ultimately using those results in air quality models.

## 3.2 Effect of particle acidity on α-pinene SOA yield

### 3.2.1 Dependence of SOA yield on particle acidity

The initial pH value of aerosol particles calculated from the E-AIM was in the range of −0.93 to −1.66 in the high- and low-$NO_x$ experiments (see Table 1). An increase of the α-pinene SOA yield with an increase of particle acidity was observed under high-$NO_x$ conditions. The final SOA yields were 5.6%, 6.6%, and 7.6% for acidic particles with the initial $NH_4/SO_4$ molar ratios of 1.0, 0.5, and 0.2, respectively (Table 1). This corresponds to 1.3, 1.6, and 1.8 times the SOA yield for neutral particles (i.e., 4.2%). Conversely, the final SOA yields for acidic particles varied from 28.6% to 36.3% under low-$NO_x$ conditions, from which a systematic increase in SOA yield with particle acidity was not observed. Clearly, the presence of acidic particles promotes the formation of α-pinene SOA under high-$NO_x$ conditions and $NO_x$ is likely involved in the acid-catalyzed reactions during α-pinene photooxidation. The dependence of α-pinene SOA yield on particle acidity under only high-$NO_x$ conditions in this study is similar to those reported by Eddingsaas et al. (2012), whereas they observed a smaller increase of SOA yield (approximately 22% compared to 30–80% here) when using acidic particles. The effect of particle

acidity on SOA formed from α-pinene has been reported to be much lower than that for isoprene, e.g., the former one was 8 times lower than the later one (Offenberg et al., 2009).

In addition to the effect of particle acidity, the α-pinene SOA yield was also possibly influenced by the liquid water content in the particles. The initial water content in the seed particles estimated by the E-AIM was on average 5.2, 6.3, and 10.3 μg m$^{-3}$ for high-NO$_x$ experiments with NH$_4$/SO$_4$ molar ratios of 1.0, 0.5, and 0.2, respectively. Therefore, more water was present in the particles with higher acidity. The higher particle water content could prompt the partitioning of gas-phase water-soluble organic species by providing a larger medium for their dissolution and therefore potentially increase the SOA

yield (Carlton and Turpin, 2013). However, there was no apparent increase in the SOA yield under low-NO$_x$ conditions, even though seed particles with similarly varied water content were used (Exp. 5–8; Table 1) and despite the fact that products with higher O/C (hence higher solubility) were formed (section 3.3). This suggests that the particle water content likely did not contribute substantially to the observed increase in α-pinene SOA yield with acidity under high-NO$_x$ conditions.

### 3.2.2  Time scale of acidity effect

SOA yield is found to be a strong function of the generated SOA mass (ΔM$_0$) (Odum et al., 1996). The time-dependent SOA yield as a function of ΔM$_0$ for acidic and neutral particles under high- and low-NO$_x$ conditions is shown in Figure 3. Under high-NO$_x$ conditions, the increase of SOA yield with particle acidity (black through green points in Figure 3a) was

much stronger in the first hour of photooxidation than in the later period, suggesting that the acidity effect was more significant in the initial period of photooxidation in this reaction system. This is possibly due to fresh acidic particles being more accessible for acid-catalyzed reactions by early α-pinene oxidation products in the initial stage. A slight decrease in the SOA yield for acidic particles was also observed after the relatively higher SOA yields within the first 30 min. A possible interpretation for such a decrease in yield is that acidic particles (i.e., the inorganic core) were gradually less accessible with

increased organic coating on acidic particles. This assumes that a phase separation of particulate organic and inorganic components occurred, from which a core-shell morphology is inferred (Drozd et al., 2013), and that the diffusion of organic molecules into the inorganic core was considerably slowed. This process was indeed possible at the studied final RH (approximately 29–43%), given that SOA could be in an amorphous solid or semisolid state with high viscosity at low to moderate RH (e.g., ≤ 30%) (Renbaum-Wolff et al., 2013; Virtanen et al., 2010). This indicates that the acidity effect is

particularly important in the initial stages of α-pinene oxidation in the presence of acidic particles. It is expected that further reactive uptake of α-pinene SOA to acidic particles might have been suppressed due to a phase separation, as has been reported by other studies (Drozd et al., 2013; Lin et al., 2014; Riva et al., 2016). The SOA yields increased nearly linearly with ΔM$_0$ after 2 hours of irradiation, suggesting that the growth of SOA mass continued after the complete consumption of the α-pinene. This is possibly due to the further oxidation of early-generation products such as carbonyls, hydrocarbonyls,

and organic nitrates, and/or the continued partitioning of gas-phase oxidation products into particle-phase. In contrast, the growth curves of SOA yields for acidic particles under low-NO$_x$ conditions were quite similar to that for neutral particles

over the irradiation time (Figure 3b), which again suggests that acidity effect is insignificant under the studied low-NO$_x$ conditions.

The acidity effect on α-pinene SOA yield was relatively strong in the first hour of irradiation under high-NO$_x$ conditions, as illustrated above. This effect was characterized more quantitatively as a function of NH$_4$/SO$_4$ molar ratio, a proxy of particle acidity, in Figure 4. Here, the SOA yields at several specific ΔM$_0$ values from 0.7 to 1.9 µg m$^{-3}$ (within the first hour in Figure 3a) are used as it represents the strongest acidity effect observed. As seen in Figure 4, the SOA yield increased nearly linearly with the decrease in the NH$_4$/SO$_4$ molar ratio. A maximum increase of 220% in SOA yield was observed for the most acidic particles (i.e., NH$_4$/SO$_4$ molar ratio = 0.2) with the ΔM$_0$ of 0.7 µg m$^{-3}$ at the irradiation time of approximately 20 min compared to those for neutral particles (NH$_4$/SO$_4$ molar ratio = 2.0) (Figure 4). This increase is much higher than the increase in the final SOA yield with particle acidity (i.e., 80% at 6 hours) in the same experiments. Furthermore, the increase in the SOA yield gradually slowed with the increase in organic mass, which is evident by the decreased trend of the slope curve derived from the fitting of SOA yield with NH$_4$/SO$_4$ molar ratios (Figure 4b). This could be again explained, at least in part, by acidic particles being less accessible over time with the coating of α-pinene SOA. Another possible cause is the consumption of sulfate due to the formation of organic sulfates (Surratt et al., 2007a, 2008). However, we cannot identify organic sulfates clearly based upon the AMS measurement, since their fragmentation results mainly in inorganic sulfate fragments (Farmer et al., 2010).

### 3.2.3 Acidity effect on later-generation SOA

Due to organic coatings on acidic particles, the effect of particle acidity on SOA formation in a later experimental stage may be underestimated when introducing seed particles before α-pinene photooxidation, in particular under low-NO$_x$ conditions with higher SOA yield. Figure 5 presents the growth curves of the generated organic aerosol mass for experiments with seed particles injected after 2 and 4 hours α-pinene photooxidation under high- and low-NO$_x$ conditions, respectively. The organic aerosol mass was normalized by the reacted α-pinene concentration before adding seed particles. Aerosol particles from the nucleation of gas molecules were not significant in these experiments (less than 50 particles cm$^{-3}$), and thus the oxidation products were likely present mainly in the gas phase prior to adding seed particles.

The generated organic aerosol mass increased immediately after adding seed particles for all experiments. This can be explained by the reactive uptake of the gas-phase α-pinene oxidation products formed in the early stages onto the acidic and ammonium sulfate seed particles. A higher increase in organic aerosol mass (up to 6 times) was observed for acidic particles than that for neutral particles in the first 2 hours under both high- and low-NO$_x$ conditions (Figure 5). This suggests that the formation and/or partitioning of organic aerosols, possibly the mixture of early- and later-generation SOA (although their proportions are unknown based on the available data), were enhanced in the presence of acidic particles even under low-NO$_x$ conditions, where no discernable acidity effect was observed previously (as seen in Figure 3b). It is postulated that this effect is apparent here since the acidic particles had not been coated previously with early-generation products of α-pinene photooxidation, which makes the acidic particles accessible to further acid-catalyzed chemistry. Eddingsaas et al. (2012) also

reported that α-pinene photooxidation products preferentially partition to highly acidic aerosols when introducing seed particles after OH oxidation under low-NO$_x$ conditions. The results in Figure 5 also indicate that later products of α-pinene oxidation were more likely to be acid-catalyzed than the early products under low-NO$_x$ conditions. Therefore, acidity effects may be different for α-pinene SOA products formed from multiple oxidation steps. A detailed analysis of those products at the molecular level is essential to fully understand this effect.

### 3.3   Chemical composition of SOA

The effect of particle acidity on the chemical composition of α-pinene SOA in high- and low-NO$_x$ experiments is examined from the distribution of organic fragments in the high-resolution organic aerosol mass spectra (see Figure S2 in the Supplement). The average fractions of organic fragment groups in the organic aerosol mass spectra for particles of different acidity are shown in Figure 6. $C_xH_y^+$ fragments (accounted for 41–44% of total signal) dominated the organic aerosol mass spectra, followed by $C_xH_yO_1^+$ (33–35%) and $C_xH_yO_2^+$ (16–17%) fragments for experiments with varied particle acidity under high-NO$_x$ conditions (Figure 6a). In contrast, $C_xH_yO_1^+$ (39–40%) was the most dominant organic fragment, followed by $C_xH_y^+$ (33–36%) and $C_xH_yO_2^+$ (17–19%) fragments under low-NO$_x$ conditions (Figure 6b). An increase in the fractions of oxygenated fragments ($C_xH_yO_1^+$ and $C_xH_yO_2^+$) and a decrease in the fraction of hydrocarbon fragments ($C_xH_y^+$) were observed under low-NO$_x$ conditions compared to those of high-NO$_x$ conditions. Also, lower O/C ratios of α-pinene SOA were observed under high-NO$_x$ conditions (0.52–0.56, averaged at the irradiation time of 1–6 hours) compared to those under low-NO$_x$ conditions (0.61–0.64, averaged at the irradiation time of 2–12 hours). This indicates that less oxygenated α-pinene SOA was formed in the presence of high NO$_x$ despite the fact that oxidants (i.e., OH and O$_3$) levels were higher during the high NO$_x$ containing experiments (see Table 1 and Figure 1).

The dependence of chemical composition and oxidation state of α-pinene SOA on NO$_x$ level is most likely associated with the different oxidation products from gas-phase chemistry of RO$_2$. For instance, peroxynitrates and organic nitrates formed from the chemical reaction of RO$_2$ and NO$_x$ are the dominant products under high-NO$_x$ conditions, whereas organic peroxides and acids formed from RO$_2$ with HO$_2$ are dominant under low-NO$_x$ conditions (Xia et al., 2008). Note that the observed variations in organic fragments in Figure 6 generally represent those over the whole photooxidation period in each experiment, since the individual mass spectrum of α-pinene SOA did not change significantly with irradiation time, as illustrated by the small standard deviations of individual fragment groups.

With the increase in particle acidity (i.e., NH$_4$/SO$_4$ molar ratio from 2.0 to 0.2) under high-NO$_x$ conditions (Figure 6a), the fractions of major fragment ions ($C_xH_y^+$ and $C_xH_yO_1^+$) decreased gradually while $C_xH_yO_2^+$ fractions increased; a slight increase in the O/C ratio from 0.52 to 0.56 was also observed. This suggests that more oxygenated SOA was possibly formed in the presence of acidic particles under high-NO$_x$ conditions. A possible interpretation is that particle acidity enhances the formation of more oxygenated SOA in particles such as larger oligomers via acid-catalyzed reactions (Gao et al., 2004), and/or promotes the partitioning of those oxidation products into particle-phase (Healy et al., 2008), or particle acidity may also help to hydrolyze unsaturated organic molecules. Conversely, there is no systematic change in the chemical composition

of α-pinene SOA with particle acidity under low-$NO_x$ conditions. Therefore, the effect of particle acidity on the chemical composition of α-pinene SOA may be important only under the studied high-$NO_x$ conditions when introducing acidic seed particles before photooxidation, which is consistent with the acidity effect on the yield of α-pinene SOA (Sect. 3.2). It is likely that acidic particles coated rapidly by earlier-generation α-pinene SOA due to the higher SOA yield under low-$NO_x$ conditions, or the reactions of $RO_2$ with $HO_2$ and $RO_2$ were dominated by termination products that were less affected by particle acidity. In addition, some oxidation products such as hydroperoxides might have reacted on the acidic particles and produced more volatile products (Liu et al., 2016), which may manifest as a decrease in the acidity effect (i.e., lower yield) for α-pinene SOA under low-$NO_x$ conditions.

### 3.4 Acidity effect on organic nitrate formation

The formation of organic nitrates from α-pinene oxidation has been reported previously in the presence of $NO_x$ (e.g., Atkinson et al., 2000; Albert et al., 2005). Nitrogen (N)-containing organic fragments ($C_xH_yN_p^+$ and $C_xH_yO_zN_p^+$) accounted for less than 10% of total organic signal in our studied conditions. These fragments were most likely contributed by organic nitrates generated from the reactions of early α-pinene oxidation intermediate ($RO_2$) with NO and $NO_2$. Organic nitrates likely account for an even higher fraction of the total organic aerosols, since their fragmentation would primarily contribute to inorganic nitrate fragments ($NO^+$ and $NO_2^+$) and other organic groups (Farmer et al., 2010). Assuming an average molecular weight of organic nitrate molecules ranging from 200 to 300 g $mol^{-1}$, where 62 g $mol^{-1}$ is attributed to the $-ONO_2$ group and the remaining from the organic mass (Boyd et al., 2015), the organic nitrate mass was estimated to be approximately 0.6–1.4 μg $m^{-3}$. This resulted in a contribution of 17.5–20.5% to total α-pinene SOA and an overall organic nitrate yield of 0.7–1.6% under high-$NO_x$ conditions in this study. Organic nitrates yield has been reported to be in the range of approximately 1% up to more than 20% for α-pinene oxidation (Aschmann et al., 2002; Nozière et al., 1999; Rindelaub et al., 2015).

Interestingly, both the fractions of $C_xH_yN_p^+$ and $C_xH_yO_zN_p^+$ fragments increased gradually with the increase in particle acidity under high-$NO_x$ conditions (Figure 6a), which is distinct from those without an apparent change under low-$NO_x$ conditions (Figure 6b). The growth curves of the total N-containing organic fragments (sum of $C_xH_yN_x^+$ and $C_xH_yO_zN_p^+$) for different acidic particles under high-$NO_x$ conditions are shown in Figure 7a. The absolute mass concentrations of total N-containing organic fragments were also enhanced with particle acidity over the irradiation period. These results indicate that organic nitrates were formed heterogeneously through a mechanism catalyzed by aerosol acidity or that acidic conditions facilitate the partitioning of gas phase nitrates into particle phase under high-$NO_x$ conditions. One possible reaction is the acid-catalyzed formation of sulfated organic nitrates through α-pinene oxidation products such as nitroxyl alcohols and carbonyls reacting with sulfuric acid (Surratt et al., 2008). Further investigations on the individual particle phase organic nitrate species at a molecular level combined with gas-particle kinetics are required to elucidate the detailed reaction mechanisms. Moreover, it has been demonstrated that acid-catalyzed hydrolysis is an important removal process for organic nitrates in the particle phase, from which organic nitrates can be converted to alcohols and nitric acid (Day et al., 2010; Hu et

al., 2011; Liu et al., 2012; Rindelaub et al., 2015). This process would also enhance the partitioning of gaseous organic nitrates into the particle phase due to the perturbation in gas/particle partitioning, and therefore decrease the organic nitrate yields both in the gas and particle phases (Rindelaub et al., 2015). The observed increase in N-containing organic fragments
with particle acidity under high-$NO_x$ conditions suggests that the production of organic nitrates generally exceeded their removal rates in this reaction system.

The time-dependent mass concentrations of $NO^+$ and $NO_2^+$ fragments for various acidic particles under high-NO conditions are shown in Figure 7b. The mass concentrations of the $NO^+$ fragment for acidic particles were higher than those of neutral particles, whereas no obvious difference in the $NO_2^+$ fragment was observed for particles with varied acidity.
Therefore, the enhanced organic nitrates by particle acidity might contribute mainly to the increase in the $NO^+$ fragment. Large relative contribution of organic nitrates to the nominal inorganic nitrate fragments is demonstrated by a higher $NO^+/NO_2^+$ ratio than those of pure ammonium nitrate (Bae et al., 2007; Boyd et al., 2015; Farmer et al., 2010; Fry et al., 2009; Xu et al., 2015b). The average $NO^+/NO_2^+$ ratio was 9.13 ± 4.24, 9.28 ± 5.19, 9.31 ± 4.27, and 10.44 ± 5.48 for particles with initial $NH_4/SO_4$ molar ratio of 2.0, 1.0, 0.5, and 0.2, respectively. These values are significantly higher than 2.6
± 0.2 from the current AMS measurement of pure ammonium nitrate, but close to that of 11 ± 8 reported for $NO_3$ oxidation of α-pinene, from which organic nitrates were likely the dominant aerosol component (Bruns et al., 2010). The increase in $NO^+/NO_2^+$ ratio with particle acidity suggests that the composition of organic nitrate species might be different under various acidic conditions, which is possibly due to the varied effect of particle acidity on the formation and/or partitioning of different organic nitrate species.

**It should be noted** that a small amount of $C_xH_yN_p^+$ and $C_xH_yO_zN_p^+$ fragments were also observed under low-$NO_x$ conditions, where NO was not added (Figure 6b). This may be contributed by the formation of minor amounts of organic nitrates from the reactions of $NO_2$ released from the chamber walls with α-pinene oxidation products. The average $NO^+/NO_2^+$ ratio was in the range of 6.92–7.91 for particles with different acidities under low-$NO_x$ conditions, which indicates that some organic nitrate species different from those under high-$NO_x$ conditions might be formed. No apparent
changes are observed in the mass fractions of $C_xH_yN_p^+$ and $C_xH_yO_zN_p^+$ fragments with particle acidity under low-$NO_x$ conditions, suggesting that acid-catalyzed formation and partitioning of those organic nitrate species were possibly insignificant.

## 4  Implications

This study investigated the effect of particle acidity on the yield and chemical composition of α-pinene SOA from
photooxidation in a photochemical reaction chamber. A nearly linear increase of α-pinene SOA yield with the increase in particle acidity was observed under high-$NO_x$ conditions, which is contrary to the insignificant acidity effect under low-$NO_x$ conditions. The potential mechanisms leading to the different acidity effects between high- and low-$NO_x$ conditions warrant further investigation. The acidity effect was relatively strong in the early photooxidation stages under high-$NO_x$ conditions, and this effect decreased gradually with the growth of SOA mass. This may be explained by a reduced accessibility of the

SOA partitioning species to the acidic particles for acid-catalyzed chemistry, possibly as a result of the SOA coating. Given that the α-pinene loading used in this study was low and the generated organic aerosol mass was relevant to ambient levels (e.g., the final ratio of organic/sulfate was 0.6–0.8 and 1.7–2.8 under high- and low-$NO_x$ conditions, respectively), similar process may also occur in the atmosphere. Consequently, an ambient acidity effect is likely stronger for newly formed particles and/or freshly formed sulfate coating. Therefore, the time scale of SOA formation with respect to acidity effects is

expected to be an important factor for field studies measuring acidity effect in the atmosphere.

      More oxygenated SOA was formed with the increase of particle acidity under high-$NO_x$ conditions. Since aerosol acidity could affect the oxidation state of aerosol particles and alter their chemical composition and other properties as demonstrated here, this may be an important process in the atmosphere and deserve further investigation. The formation of SOA from later-generation gas phase products was enhanced by particle acidity even under low-$NO_x$ conditions when

introducing acidic seed particles after α-pinene photooxidation. This suggests that the overall acidity effect on the formation of SOA could be underestimated, and that more systematic studies are necessary to evaluate the acidity effect on SOA generated from multiple oxidation steps. This effect could also be important in the atmosphere under conditions where α-pinene oxidation products in the gas-phase originating in forested areas (with low $NO_x$ and $SO_x$) are transported to regions abundant in acidic aerosols such as power plant plumes or urban regions. Organic nitrates in these experiments may be

formed heterogeneously through a mechanism catalyzed by particle acidity and/or the acidic conditions facilitate the partitioning of gas phase nitrates into the particle phase under high-$NO_x$ conditions. This implies that aerosol acidity could also be of importance in the atmosphere by altering the deposition patterns and rates of gas phase $NO_x$ via its conversion to particle nitrates with differing atmospheric lifetimes.

      Despite the initial pH value of aerosol particles investigated in this study (−0.93 to −1.72) being in the higher acidity

range relative to that generally observed for ambient aerosols, pH values less than −2.0 have been reported for atmospheric aerosol particles and haze droplets (Herrmann et al., 2015). It is therefore expected that the effect of particle acidity observed in this study is relevant to the ambient atmosphere, especially in regions enriched with acidic aerosols, and possibly during initial particle growth via sulfuric acid. Moreover, we have studied the acidity effect under more realistic RH conditions. While RH is an important factor affecting the concentrations of [$H^+$], the kinetics of hydrolysis reactions, and the physical

properties of SOA such as viscosity, more investigation over a broader RH range are essential to understand the acidity effect in the real atmosphere. Finally, further studies on SOA formation from various other hydrocarbons under conditions near ambient atmospheric levels will be valuable in understanding the complex physical and chemical interactions facilitated by aerosol acidity and evaluating the acidity effect more accurately, and to ultimately incorporate such effects into regional air quality model for improved SOA prediction.

## Acknowledgments

This study was funded by the Joint Oil Sands Monitoring Program between Alberta Environment and Sustainable Resource Development and Environment and Climate Change Canada.

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

**Tables**

**Table 1. Experimental conditions and SOA yields from OH-initiated photooxidation of α-pinene under high- and low-NO$_x$ conditions.**

| Exp. | NH$_4$/SO$_4$ ratio [a] | Initial seed composition, [b] molality (mole kg$^{-1}$) | Aerosol pH [c] | Temp. [d] (°C) | RH [e] (%) | Seed (µg m$^{-3}$) | NO (ppb) | α-pinene (ppb) | ΔHC (µg m$^{-3}$) | ΔM$_0$ (µg m$^{-3}$) | Yield (%) |
|---|---|---|---|---|---|---|---|---|---|---|---|
| **High-NO$_x$ conditions** | | | | | | | | | | | |
| 1 | 2 | (NH$_4$)$_2$SO$_4$ (no liquid phase) | | 24–31 | 47–29 | 4.4 | 66 | 15.9 | 84.2 | 3.5 | 4.2±0.1 |
| 2 | 1 | H$^+$=3.2, NH$_4^+$=15.3, HSO$_4^-$=11.2, SO$_4^{2-}$=3.7 | −1.31 | 23–30 | 58–34 | 8.4 | 69 | 17.6 | 93.4 | 5.2 | 5.6±0.1 |
| 3 | 0.5 | H$^+$=5.0, NH$_4^+$=7.0, HSO$_4^-$=7.5, SO$_4^{2-}$=2.3 | −1.50 | 24–30 | 61–38 | 6.3 | 68 | 13.6 | 71.8 | 4.7 | 6.6±0.1 |
| 4 | 0.2 | H$^+$=7.2, NH$_4^+$=2.7, HSO$_4^-$=6.2, SO$_4^{2-}$=1.9 | −1.66 | 26–34 | 58–33 | 7.9 | 72 | 17.0 | 89.9 | 6.8 | 7.6±0.2 |
| **Low-NO$_x$ conditions** | | | | | | | | | | | |
| 5 | 2 | (NH$_4$)$_2$SO$_4$ (no liquid phase) | | 25–32 | 67–43 | 12.6 | <0.3 | 19.6 | 96.7 | 34.1 | 35.2±1.1 |
| 6 | 1 | H$^+$=1.9, NH$_4^+$=14.8, HSO$_4^-$=8.5, SO$_4^{2-}$=4.1 | −0.93 | 25–32 | 64–37 | 12.6 | <0.3 | 17.4 | 79.1 | 22.6 | 28.6±1.5 |
| 7 | 0.5 | H$^+$=3.2, NH$_4^+$=10.6, HSO$_4^-$=8.0, SO$_4^{2-}$=2.9 | −1.22 | 26–33 | 64–38 | 11.9 | <0.3 | 19.3 | 92.6 | 33.7 | 36.3±1.5 |
| 8 | 0.2 | H$^+$=5.3, NH$_4^+$=4.1, HSO$_4^-$=5.6, SO$_4^{2-}$=1.9 | −1.35 | 24–33 | 66–36 | 11.4 | <0.3 | 19.5 | 88.6 | 28.4 | 32.0±1.9 |
| **Adding seeds after photooxidation** | | | | | | | | | | | |
| 9 | 2 | (NH$_4$)$_2$SO$_4$ (no liquid phase) | | 23–31 | 57–34 | 9.7 | 82 | 20.4 | | | |
| 10 | 0.5 | H$^+$=5.9, NH$_4^+$=7.0, HSO$_4^-$=8.4, SO$_4^{2-}$=2.3 | −1.72 | 24–31 | 56–33 | 12.2 | 72 | 18.5 | | | |
| 11 | 2 | (NH$_4$)$_2$SO$_4$ (no liquid phase) | | 25–33 | 68–42 | 7.4 | <0.3 | 16.1 | | | |
| 12 | 0.5 | H$^+$=4.9, NH$_4^+$=9.6, HSO$_4^-$=9.3, SO$_4^{2-}$=2.6 | −1.64 | 25–33 | 57–33 | 11.4 | <0.3 | 17.3 | | | |

[a] NH$_4$/SO$_4$ molar ratios of ammonium sulfate/sulfuric acid aqueous solution used for atomizing seed particles. [b] Initial seed composition was estimated using the E-AIM II. [c] Aerosol pH was calculated with the E-AIM output. [d] Initial and final temperature inside the chamber. [e] Initial and final RH inside the chamber.

**Table 2. Comparison of experimental parameters and SOA yields reported in literature for the photooxidation of α-pinene.**

| Reference | Temp. (°C) | RH (%) | Oxidant | Seed | $NO_x$ (ppb) | α-pinene (ppb) | ΔHC (µg m$^{-3}$) | ΔM$_0$ (µg m$^{-3}$) | SOA Yield (%) |
|---|---|---|---|---|---|---|---|---|---|
| Chu et al. (2014) | 28 | 12, 50 | HONO | AS | n.a. | 8.1, 11.7 | n.a. | 5.9, 9.3 | 15.1, 23.4 |
| | 28 | 12,50 | HONO | FeSO$_4$ | n.a. | 9.7, 10.0 | n.a. | 5.0, 2.9 | 10.9, 5.7 |
| Kim and Paulson (2013) | 33–42 [a] | 15–25 [a] | propene | no seed | 47–230 | 143–153 | n.a. | 9–118 | 5.9–17 |
| Eddingsaas et al. (2012) | 20–25 | <10 | H$_2$O$_2$ | AS | n.a. | 45.0–48.5 | 247–265 | 63.5–76.6 | 25.7–28.9 |
| | 20–23 | <10 | HONO, CH$_3$ONO | AS | ~800 | 44.9–52.4 | 249–258 | 37.2–60.3 | 14.4–24.2 |
| Ng et al. (2007a) | 23–25 | 5.3–6.4 | H$_2$O$_2$ | AS | 0,1 | n.a. | 76.7, 264.1 | 29.3, 121.3 | 37.9–45.8 |
| | 25–26 | 3.3–3.7 | H$_2$O$_2$+NO, HONO | AS | 198–968 | n.a. | 69.8–259.1 | 4.5–40.8 | 6.6–21.2 |
| Kleindienst et al. (2006) | 26.3 | 29 | NO$_x$ | no seed | 242, 543 | 2550 | 1190, 815 | 130, 67.3 [b] | 10.9, 8.3 [c] |
| | 26.3 | 29 | NO$_x$ | sulfate | 242, 543 | 2550 | 1190, 815 | 87–172 [b] | 10.7–14.5 [c] |
| Takekawa et al. (2003) | 10 | ~60 | propene | Na$_2$SO$_4$ | 30–53 | 55–100 | 260–540 | 36–89 | 20–23 |
| | 30 | ~60 | propene | Na$_2$SO$_4$ | 54–102 | 93–196 | 500–1000 | 20–95 | 5.2–10 |
| Odum et al. (1996) | 35–40 | ~10 | propene | AS | 300 | ~19–143 | 104–769 | 1.3–96.0 | 1.25–12.5 |

[a] Final temperature and RH were presented. [b] OC mass was reported. [c] SOC yield was reported.

**Figures**

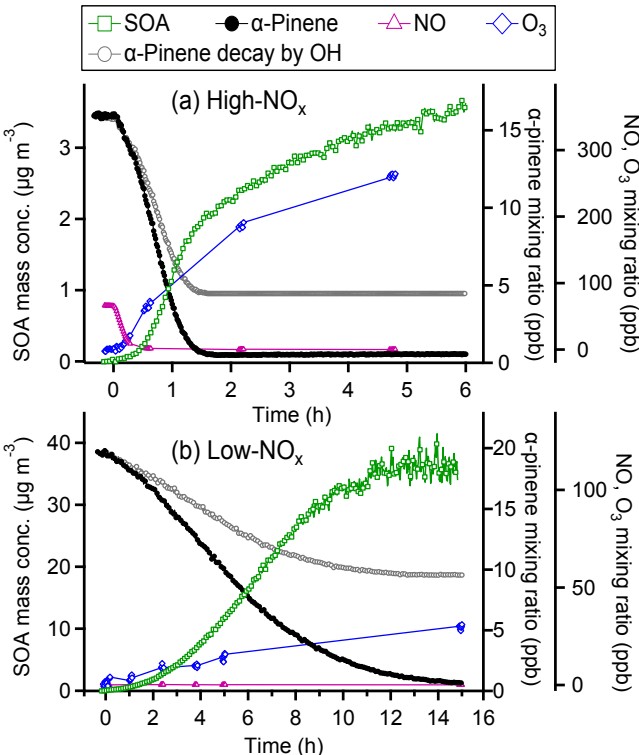

**Figure 1. Time series of the mass concentrations of generated SOA and the mixing ratios of NO, O₃, total α-pinene decay, and OH**
**consumed α-pinene in (a) high- and (b) low-NOₓ experiments using ammonium sulfate as seed particles. Time = 0 hour is defined as α-pinene photooxidation initiated when lamps were turned on. The presented SOA mass concentrations have been corrected for particle wall loss according to the decay of sulfate mass.**

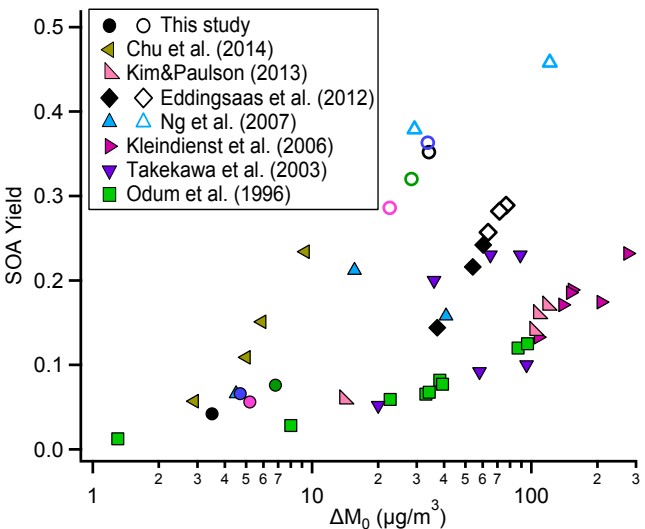

Figure 2. Comparison of final α-pinene SOA yields as a function of organic mass concentration ($\Delta M_0$) under high- and low-$NO_x$ conditions in this study with those reported in literature. The solid and open symbols represent the SOA yields under high- and low-$NO_x$ conditions, respectively. The black, pink, blue, and green cycles represent the SOA yield for experiments in this study with $NH_4/SO_4$ molar ratios of 2.0, 1.0, 0.5, and 0.2, respectively. A factor of 1.6 was used to convert SOC yield and OC mass concentration to SOA yield and OA mass concentration in Kleindienst et al. (2006).

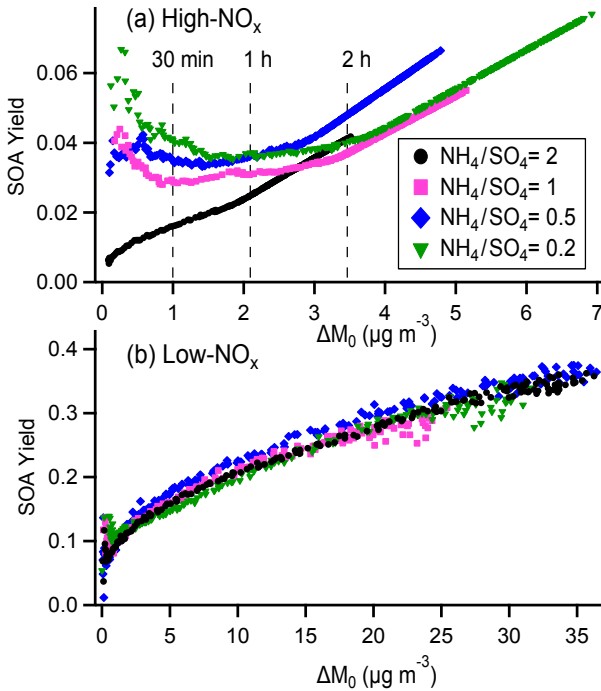

**Figure 3. SOA yields as a function of organic mass concentrations for experiments using seed particles with varied acidity levels under (a) high- and (b) low-$NO_x$ conditions. The dashed lines in (a) represent the irradiation time at approximately 30 min, 1 hour, and 2 hours, respectively.**


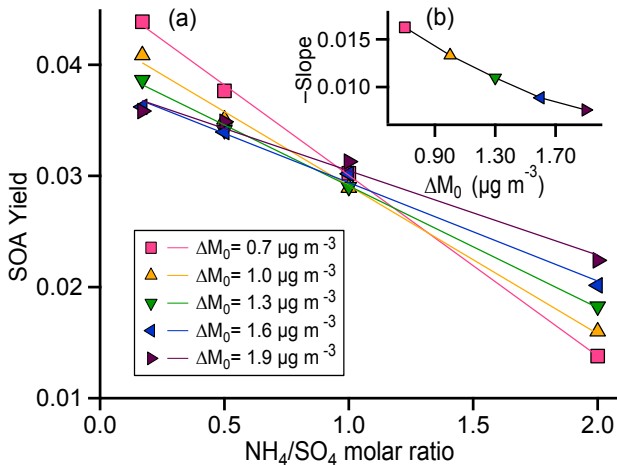

**Figure 4. (a) SOA yield versus NH$_4$/SO$_4$ molar ratio in the initial period of photooxidation (approximately 0–1 hours) under high-NO$_x$ conditions. The colored lines represent the linear fitting of the markers. The SOA yields at specific $\Delta M_0$ values were retrieved from the plotting of SOA yields versus $\Delta M_0$ in Figure 3a. (b) The negative slope derived from the fitting of SOA yields with NH$_4$/SO$_4$ molar ratios in (a) decreased with $\Delta M_0$.**

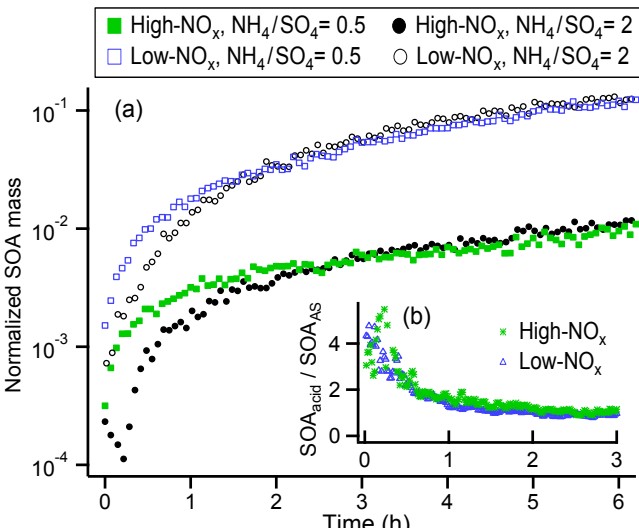

**Figure 5. (a) The increase of SOA mass with time for experiments injecting ammonium sulfate and acidic seed particles after α-pinene photooxidation for 2 and 4 hours under high- and low-NOₓ conditions, respectively (Exp. 9–12 in Table 1). Time= 0 hour represents the beginning of reactive uptake of oxidation products after seed particles were added. The SOA mass was normalized by the reacted α-pinene concentration before adding seed particles. (b) The ratio of SOA mass for acidic particles to that of ammonium sulfate particles.**


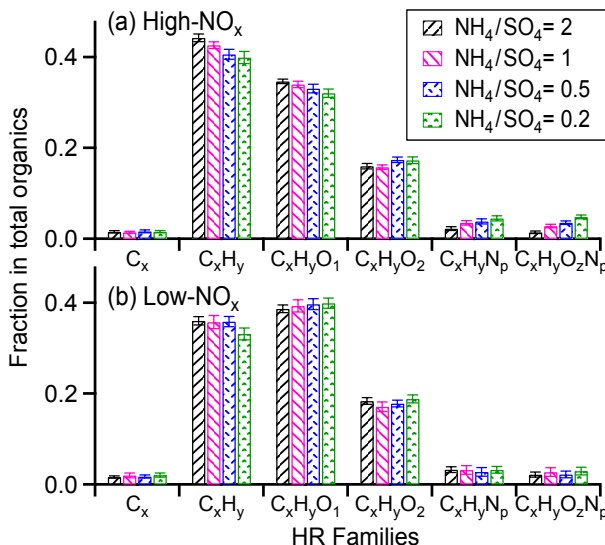

**Figure 6. The mass fractions of organic fragment groups in total organic aerosols under (a) high- and (b) low-NO$_x$ conditions. The organic mass spectra were averaged for the irradiation times of 1–6 and 2–12 hours under high- and low-NO$_x$ conditions, respectively. The bars represent the standard deviations (± 1σ) of the mean values for individual fragment groups.**

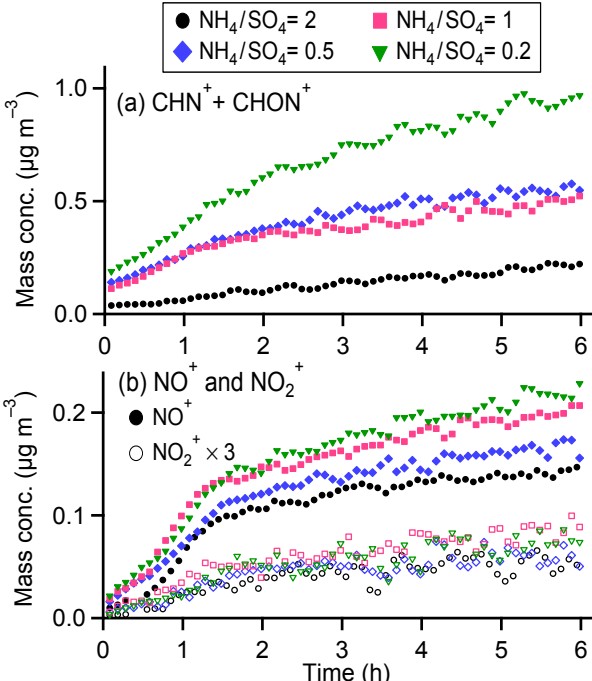

**Figure 7. The temporal variations of (a) total N-containing organic fragments (the sum of $C_xH_yN_z^+$ and $C_xH_yO_zN_p^+$) and (b) $NO^+$ and $NO_2^+$ fragments for experiments using ammonium sulfate and acidic particles under high-$NO_x$ conditions.**