# Peer review of "The effect of particle acidity on secondary organic aerosol formation from α-pinene photooxidation under atmospherically relevant conditions"

_Atmospheric Chemistry and Physics, 2016_

## Referee Comment (RC1) · Anonymous Referee #1 · 22 Apr 2016

Comments on '*The effect of particle acidity on secondary organic aerosol formation from α-pinene photooxidation under atmospherically relevant conditions*'

This work presents laboratory chamber studies on the effect of particle acidity on the secondary organic aerosols (SOA) formation from OH-initiated oxidation of α-pinene, a topic that has received much attention yet still under debate. The authors concentrate their discussions on how seed particle acidity would affect the SOA yield and elemental composition during different stages of photooxidation and under different reaction regimes (high vs. low $NO_x$). Overall, these experimental observations would certainly contribute a relatively complete dataset to studies that highlight the role of aerosol acidity in SOA production and aging. However, the major conclusions are potentially biased by the experimental protocols, in particular, the interference of $NO_2$ under low $NO_x$ conditions as well as liquid water contents under different particle acidities. The authors are suggested to conduct additional experiments and perform modeling calculations to strengthen their conclusions prior to consideration of being published on *Atmospheric Chemistry and Physics*.

**General Comments**

1. Low vs. High $NO_x$ conditions

The authors observed enhanced nitrogen-containing organic fragments formation with increasing aerosol acidity under high $NO_x$ conditions. However, the fraction of these fragments remains constant at different particle acidities under low $NO_x$ conditions, as shown in Figure 6. By examination of the ozone profile shown in Figure 1, one realizes that moderate ozone production is still occurring under the so-called 'low $NO_x$ conditions', owing to the photolysis of $NO_2$ that are either released from the chamber wall or penetrated from the enclosure air, which is OK and cannot be completely avoided in most chamber setups. The production of ozone indicates a rapid recycling of the NO-$NO_2$ chemistry, and thus the active role of NO in the production of the nitrogen-containing organic fragments via the $RO_2$+NO pathway under 'low $NO_x$ conditions'. As

shown in Figure 6, the fraction of these nitrogen-containing organic fragments under low $NO_x$ conditions seems comparable with those under high $NO_x$ conditions, indicating that they cannot be simply artifacts in the AMS measurement. But the question is why the dependence of these nitrogen-containing organic fragments on particle acidity is only observed under high $NO_x$ conditions? The authors need to give a convincing explanation on this inconsistency.

Another issue is that the so-called 'high $NO_x$ condition' does not necessarily lead to the $RO_2+NO$ reaction dominant regime over the entire course of the experiment. As shown in Figure 1, NO concentration seems to be completely depleted after a few min of reaction, which is expected due to the intense formation of nitric acid and its rapid deposition on the wall and particles. The authors are suggested to carry out continuous NO injection after the initial consumption of NO to truly achieve the 'high $NO_x$ condition'. Alternatively, HONO or $CH_3ONO$ can be used as the OH precursor under high $NO_x$ conditions.

2. First vs. later generation SOA products

The strong dependence of SOA yield on particle acidities under high $NO_x$ conditions was only observed during the initial photooxidation stage, where the total SOA mass concentration is less than 2 ug/m$^3$. As the reaction proceeds, SOA yields are eventually comparable at different particle acidities and the authors explained this phenomenon as the unavailability of free $H^+$ due to thick organic coatings. Note that this argument is established under the assumption that the core of the particles are (semi)-solid and there is no particle-phase diffusion over the entire course of several hours of reaction, which is doubtful considering the moderate RH conditions employed in the experiments. One suspects that if the observed SOA yield dependence at the initial particle growth period on the acidity can be simply attributed to the uncertainties in the AMS measurement when the overall organic loadings are extremely low (< 2 ug/m$^3$). The authors are suggested to perform additional calibration experiments:

1) Prepare a mixture of pinonic acid (a surrogate for α-pinene photo-oxidation products) and ammonium sulfate salt under different acidities, atomize the mixtures into the chamber under controlled RH and T conditions, and sample

the suspended particles by AMS. The authors need to verify if AMS measured organic loadings are identical under different seed acidities even at low organic levels. This calibration experiments can also be used as a means to correct the AMS collection efficiency, if different from the default value used.

2) Alternatively, the authors could repeat experiments # 9-12, but adding seed particles after only 30 min of reaction. Would the dependence of SOA yield on the organic loadings still be observed?

3. Aerosol liquid water content

The hygroscopic growth curves for ammonium sulfate seed at different acidities are significantly different. At ~ 50% RH, diameter changes of $(NH_4)_2SO_4$ and $NH_4HSO_4$ are 0% and 20% at equilibrium, respectively, due to water uptake (Seinfeld and Pandis, 2008). For the experimental conditions in this study, i.e., dry seed particles with different acidities at 50-70% RH, the water content would vary significantly from zero to several tens of percentage in mass, depending the amount of sulfuric acid in the particle phase. As a result, the observed 'acidity effect', if any, should really be the coupled effect of both acidities and water content in aerosols. Another concern is the different phase states of particles could potentially lead to different bouncing effect on the AMS vaporizer, thus changing the collection efficiency for different seed particles. In view of these uncertainties, the authors are suggested to conduct a series of experiments using hydrated seed particles to begin with.

4. Particle acidity and atmospheric relevant conditions

In view of the main focus of this study, the seed particle acidities need to be estimated based on the AMS measured inorganic composition and thermodynamic models such as E-AIM (http://www.aim.env.uea.ac.uk/aim/aim.php). The authors are also suggested to discuss how relevant the acidities used in the chamber experiments with atmospheric conditions.

**Minor Comments**

1. Page 2, Line 50: Ester dimers are observed as an important class of products from the ozonolysis of $\alpha$-pinene (e.g., Zhang et al., PNAS, 2015). Epoxides are certainly produced, but from a minor pathway (e.g., Eddingsaas et al., ACP, 2012), and ring-opening reactions proceed rapidly following the reactive uptake of epoxides.

2. Page 2, Line 57: It has been well recognized that sulfate esters contribute to a large fraction of SOA mass via reactive uptake of IEPOX, a second generation oxidation products from isoprene photooxidation under low NO conditions, onto acidified aerosols. However, the exact mass fraction of sulfate esters in the $\alpha$-pinene system has not yet been determined. Surratt et al. (2007) and (2008) report for the first time the evidence of organosulfate in $\alpha$-pinene derived SOA, but not quantification.

3. Page 5, Line 132: Have the authors performed experiments monitoring the decay of pure sulfate particles? How was the decay rate compared with particles coated with organics?

4. Page 5, Line 134: How was the 0.1 ug/cm$^{-3}$ mass from self-nucleation estimated?

5. Page 5, Line 139: What is the estimated $NO_3$ photolysis rate?

---

## Referee Comment (RC2) · Anonymous Referee #2 · 27 Apr 2016

Review of "The effect of particle acidity on secondary organic aerosol formation from α-pinene photooxidation under atmospherically relevant conditions" by Han et al.

The authors performed laboratory chamber experiments to study the effect of particle acidity on α-pinene SOA. Firstly, the authors found that the particle acidity has small effect on α-pinene SOA yield under low-$NO_x$ conditions, but has large effects on α-pinene SOA yield under high-$NO_x$ conditions. This has been shown in Eddingsaas et al. (2012a). Secondly, the authors showed that α-pinene SOA formation under low-$NO_x$ conditions is influenced if the particle seed is injected after α-pinene photooxidation. This has also been shown in Eddingsaas et al. (2012a). Thirdly, the authors observed that the fraction of $C_xH_yN_z^+$ and $C_xH_yO_zN_p^+$ fragments in total organics increases with particle acidity, which is the only new finding in the manuscript. Considering the lack of novel findings in the manuscript, I would not recommend this manuscript for publication in its current state.

Major comments:

1. This manuscript does not represent a substantial contribution to scientific progress, because most of the findings have been shown in Eddingsaas et al. (2012a). The authors should try to differentiate this study from Eddingsaas et al. (2012a). For example, while the RH was below 10% in Eddingsaas et al. (2012a), the experiments were conducted under humid conditions in this study. Thus, discussions on dry vs. humid could be added. In addition, the authors should provide more insights into why the effects of acidic seed are different between low-$NO_x$ and high-$NO_x$ conditions.

2. The authors observed that the mass fraction of $C_xH_yN_z^+$ and $C_xH_yO_zN_p^+$ in total OA increases with particle acidity under high-$NO_x$ conditions. However, the discussions on this observation are highly speculative.

(1) In order to explain this observation, the authors propose "organic nitrates may be formed heterogeneously through a mechanism catalyzed by particle acidity". line 191-192, the authors suggest that particle acidity can facilitate the gas phase $RO_2$ and $NO_x$ reaction and organic nitrate

formation. This mechanism is highly speculative. The authors need to provide more evidence and cite related reference to support the proposed mechanism.

(2) line 322-323. The change in $NO^+/NO_2^+$ ratio is a reflection of the change in organic nitrate composition, instead of organic nitrate amount. The increasing in $NO^+/NO_2^+$ ratio with particle acidity likely suggests that particle acidity has different effects on the partition of different organic nitrate species. This is one possible explanation for the observation that the mass fraction of organic nitrates increases with particle acidity under high-$NO_x$ conditions.

(3) As shown in figure 6b, the mass fractions of $C_xH_yN_z^+$ and $C_xH_yO_zN_p^+$ under low-$NO_x$ conditions are similar to that under high-$NO_x$ conditions. How are organic nitrates formed under low-$NO_x$ conditions? Why are the mass fractions of $C_xH_yN_z^+$ and $C_xH_yO_zN_p^+$ under low-$NO_x$ conditions not affected by particle acidity?

(4) line 296. The organic nitrate yield reported in this study is misleading. It is because the fragmentation of organic nitrate in AMS give rise to $C_xH_yO_z^+$, which accounts for a large mass fraction in organic nitrate but not included in the yield calculation in this study. The authors could estimate the organic nitrate yield based on the concentration of the sum of $NO^+ + NO_2^+$ and assumed molecular weight of organic nitrate (A.W.Rollins, 2012; Boyd et al., 2015; Xu et al., 2015a). Also, are $NO^+$ and $NO_2^+$ included in the SOA yield and O:C calculation?

3. Many results are confusing.

(1) Figure 3: the SOA yield curves under both conditions are not typical and problematic (Griffin et al., 1999). Under low-$NO_x$ conditions, why would SOA yield decrease with delta_Mo at the beginning of the experiments? Under high-$NO_x$ conditions, what causes the SOA yield decrease within the first 30min? If one looks at the figure 1, the yield curve is expected to be monotonic. Is the weird yield curve caused by the SOA wall loss correction? What does the sulfate concentration (measured by AMS) look like over the course of the experiments? Does the sulfate concentration (measured by AMS) change once organics are formed?

(2) Figure 4: This figure is confusing. Firstly, following the same delta_Mo, the authors are comparing the SOA yield under different $NH_4/SO_4$ ratio. The trend shown in figure 4 does not hold if the authors add data points when delta_Mo = ~0.25 ug/m$^3$ (where SOA yield peaks for

NH$_4$/SO$_4$ = 0.2). Secondly, many data points are obtained from the period when SOA yield decreases with delta_Mo, the reason for which is not clear yet.

(3) Figure 5: Firstly, the y-axis scale is misleading. Although it seems that SOA yield increases a lot once injecting seed, the actual enhancement is only on the order of 0.01, which is within measurement uncertainty. Secondly, is there organics associated with sulfate seed in the atomizing solution? Based on some tests in our lab, the injection of sulfate would introduce organics, which comes from the atomizing solution, even if HPLC-grade DI water is used to make solution. Actually, the organics associated with sulfate seed may explain the immediate OA increase after adding seed particles (line 239). Thirdly, how is the increase of SOA yields calculated? What's the reference? Fourthly, the results are not consistent with Eddingsaas et al. (2012a) (figure 7), who showed that AS particles have no effect on for both low and high NO$_x$ conditions.

Minor comments:

1. Table 1: (1) Since both H$^+$ and LWC are modeled by E-AIM in the study, I suggest the authors to replace NH$_4$/SO$_4$ by particle pH, because NH$_4$/SO$_4$ or ion balance is not a good proxy for particle pH (Guo et al., 2015; Hennigan et al., 2015). (2) Are the [NH$_4$] and [SO$_4$] input for E-AIM obtained from aqueous solution or AMS measurements? The latter should be used, because the NH$_3$/NH$_4^+$ partitioning would cause the real [NH$_4$]/[SO$_4$] ratio different from that in aqueous solution.

2. Figure 1: (1) The legend "OH consumed a-pinene" is confusing because cumulative a-pinene consumed by OH should increase with time instead of decreasing as shown in the figure. I suggest to change the y-axis to "Δa-pinene consumed by OH". (2) Is SOA mass concentration shown in this figure corrected for wall loss?

3. line 15. The core-shell model contradicts with literature. For example, Renbaum-Wolff et al. (2013) measured the viscosity of a-pinene SOA and calculated that the mixing time is on the order of 10-100s under 40-50% RH.

4. line 18-20. The authors found that the SOA is more oxidized under low-NO$_x$ conditions than high-NO$_x$ conditions. However, Chhabra et al. (2011) figure 2c showed that the O:C of a-pinene SOA under low-NO$_x$ (using H$_2$O$_2$) and high-NO$_x$ conditions (using CH$_3$ONO) are similar.

5. line 35. It should be "enhance the reactive uptake of gas phase organics", instead of "particle phase".

6. line 37. Cite Surratt et al. (2010) and Xu et al. (2015a), who showed the effect of sulfate on the reactive uptake of IEPOX.

7. line 43. It is not accurate to state that atmospheric chemistry models do not consider the dependence of SOA formation on aerosol acidity. Large efforts have been devoted to consider the effect of particle acidity for SOA through IEPOX uptake (Marais et al., 2016; McNeill et al., 2012; Pye et al., 2013). Please rephrase this sentence.

8. line 50-52 and line 63-66. I don't agree with the authors that "large discrepancies among experiments remain with respect to the effects of aerosol acidity on SOA formation". The seemingly "contradictory" observations among studies listed in the manuscript are just due to the difference in experimental conditions. For example, the effects of particle acidity on $\alpha$-pinene SOA formation are different for low-$NO_x$ and high-$NO_x$ conditions. Thus, the previous studies cited in the manuscript only show the complexity of this scientific question, instead of the discrepancy.

9. line 124. The collection efficiency of AMS. Was a dryer deployed upstream of AMS and SMPS? If not, considering the high RH in this study, the particle water could affect the comparison between AMS and SMPS.

10. line 133-134. How do the authors estimate the concentration of organics from self-nucleation? Is the particle size distribution bimodal? Please show the particle size distribution measured by SMPS.

11. line 166-167. The lower SOA yield under high-$NO_x$ conditions is due to $RO_2$ reacts with NO and likely undergo fragmentation to produce volatile species, not due to the formation of organic nitrates. According to group contribution method by Pankow and Asher (2008), the reduction in vapor pressure by adding of one nitrate functional group is similar to that of adding one hydroxyl group.

12. line 246-247. Eddingsaas et al. (2012a) is not properly cited. Section 3.3 in Eddingsaas et al. (2012a) stated that "Under high-$NO_2$ conditions, no additional SOA is formed after the addition of either neutral or acidic seed particles in the dark". Thus, the finding in this study is not consistent with Eddingsaas et al. (2012a).

13. line 249-250. The authors propose that the early generation products don't participate in acid catalysis. This is not consistent with Eddingsaas et al. (2012a) (figure 8), who showed that the first generation products can partition to acidic particles.

14. line 267-270. The authors need to cite previous studies which discussed the gas phase products from a-pinene oxidation. Especially, Eddingsaas et al. (2012b) showed that pinonaldehyde is important intermediate under both low and high-NO$_x$ conditions.

15. line 270-272. Xu et al. (2014) is not properly cited. Instead of showing that isoprene SOA is more oxidized under low-NO$_x$ conditions, Xu et al. (2014) showed that the oxidation state of isoprene SOA shows a non-linear dependence on NO$_x$ level.

16. line 318. Please cite Boyd et al. (2015) and Xu et al. (2015b).

17. line 331. What's the Org/SO$_4$ ratio in this study? Is it atmospherically relevant?

18. line 342. The authors have already ruled out the mechanism "acidic seed facilitates the partitioning of gas phase organic nitrate" (line 308-310). Please rephrase this sentence.

19. line 495. The author list of this citation is wrong.

References

A.W.Rollins: Evidence for NOx Control over Nighttime SOA Formation, science, 337, 2012.

Boyd, C. M., Sanchez, J., Xu, L., Eugene, A. J., Nah, T., Tuet, W. Y., Guzman, M. I., and Ng, N. L.: Secondary organic aerosol formation from the β-pinene+NO3 system: effect of humidity and peroxy radical fate, Atmos. Chem. Phys., 15, 7497-7522, 10.5194/acp-15-7497-2015, 2015.

Chhabra, P. S., Ng, N. L., Canagaratna, M. R., Corrigan, A. L., Russell, L. M., Worsnop, D. R., Flagan, R. C., and Seinfeld, J. H.: Elemental composition and oxidation of chamber organic aerosol, Atmos Chem Phys, 11, 8827-8845, DOI 10.5194/acp-11-8827-2011, 2011.

Eddingsaas, N. C., Loza, C. L., Yee, L. D., Chan, M., Schilling, K. A., Chhabra, P. S., Seinfeld, J. H., and Wennberg, P. O.: α-pinene photooxidation under controlled chemical conditions – Part 2: SOA yield and composition in low- and high-NOx environments, Atmos. Chem. Phys., 12, 7413-7427, 10.5194/acp-12-7413-2012, 2012a.

Eddingsaas, N. C., Loza, C. L., Yee, L. D., Seinfeld, J. H., and Wennberg, P. O.: α-pinene photooxidation under controlled chemical conditions – Part 1: Gas-phase composition in low- and

high-NO$_x$ environments, Atmos. Chem. Phys., 12, 6489-6504, 10.5194/acp-12-6489-2012, 2012b.

Griffin, R. J., Cocker, D. R., Flagan, R. C., and Seinfeld, J. H.: Organic aerosol formation from the oxidation of biogenic hydrocarbons, J Geophys Res-Atmos, 104, 3555-3567, Doi 10.1029/1998jd100049, 1999.

Guo, H., Xu, L., Bougiatioti, A., Cerully, K. M., Capps, S. L., Hite Jr, J. R., Carlton, A. G., Lee, S. H., Bergin, M. H., Ng, N. L., Nenes, A., and Weber, R. J.: Fine-particle water and pH in the southeastern United States, Atmos. Chem. Phys., 15, 5211-5228, 10.5194/acp-15-5211-2015, 2015.

Hennigan, C. J., Izumi, J., Sullivan, A. P., Weber, R. J., and Nenes, A.: A critical evaluation of proxy methods used to estimate the acidity of atmospheric particles, Atmos. Chem. Phys., 15, 2775-2790, 10.5194/acp-15-2775-2015, 2015.

Marais, E. A., Jacob, D. J., Jimenez, J. L., Campuzano-Jost, P., Day, D. A., Hu, W., Krechmer, J., Zhu, L., Kim, P. S., Miller, C. C., Fisher, J. A., Travis, K., Yu, K., Hanisco, T. F., Wolfe, G. M., Arkinson, H. L., Pye, H. O. T., Froyd, K. D., Liao, J., and McNeill, V. F.: Aqueous-phase mechanism for secondary organic aerosol formation from isoprene: application to the southeast United States and co-benefit of SO2 emission controls, Atmos. Chem. Phys., 16, 1603-1618, 10.5194/acp-16-1603-2016, 2016.

McNeill, V. F., Woo, J. L., Kim, D. D., Schwier, A. N., Wannell, N. J., Sumner, A. J., and Barakat, J. M.: Aqueous-Phase Secondary Organic Aerosol and Organosulfate Formation in Atmospheric Aerosols: A Modeling Study, Environ Sci Technol, 46, 8075-8081, 10.1021/es3002986, 2012.

Pankow, J. F., and Asher, W. E.: SIMPOL.1: a simple group contribution method for predicting vapor pressures and enthalpies of vaporization of multifunctional organic compounds, Atmos Chem Phys, 8, 2773-2796, 2008.

Pye, Pinder, R. W., Piletic, I. R., Xie, Y., Capps, S. L., Lin, Y. H., Surratt, J. D., Zhang, Z. F., Gold, A., Luecken, D. J., Hutzell, W. T., Jaoui, M., Offenberg, J. H., Kleindienst, T. E., Lewandowski, M., and Edney, E. O.: Epoxide Pathways Improve Model Predictions of Isoprene Markers and Reveal Key Role of Acidity in Aerosol Formation, Environ Sci Technol, 47, 11056-11064, Doi 10.1021/Es402106h, 2013.

Renbaum-Wolff, L., Grayson, J. W., Bateman, A. P., Kuwata, M., Sellier, M., Murray, B. J., Shilling, J. E., Martin, S. T., and Bertram, A. K.: Viscosity of α-pinene secondary organic material and implications for particle growth and reactivity, Proceedings of the National Academy of Sciences, 110, 8014-8019, 10.1073/pnas.1219548110, 2013.

Surratt, J. D., Chan, A. W. H., Eddingsaas, N. C., Chan, M. N., Loza, C. L., Kwan, A. J., Hersey, S. P., Flagan, R. C., Wennberg, P. O., and Seinfeld, J. H.: Reactive intermediates revealed in secondary organic aerosol formation from isoprene, P Natl Acad Sci USA, 107, 6640-6645, DOI 10.1073/pnas.0911114107, 2010.

Xu, L., Kollman, M. S., Song, C., Shilling, J. E., and Ng, N. L.: Effects of NOx on the Volatility of Secondary Organic Aerosol from Isoprene Photooxidation, Environ Sci Technol, 48, 2253-2262, 10.1021/es404842g, 2014.

Xu, L., Guo, H., Boyd, C. M., Klein, M., Bougiatioti, A., Cerully, K. M., Hite, J. R., Isaacman-VanWertz, G., Kreisberg, N. M., Knote, C., Olson, K., Koss, A., Goldstein, A. H., Hering, S. V., de Gouw, J., Baumann, K., Lee, S.-H., Nenes, A., Weber, R. J., and Ng, N. L.: Effects of anthropogenic emissions on aerosol formation from isoprene and monoterpenes in the southeastern United States, Proceedings of the National Academy of Sciences, 112, 37-42, 10.1073/pnas.1417609112, 2015a.

Xu, L., Suresh, S., Guo, H., Weber, R. J., and Ng, N. L.: Aerosol characterization over the southeastern United States using high-resolution aerosol mass spectrometry: spatial and seasonal variation of aerosol composition and sources with a focus on organic nitrates, Atmos. Chem. Phys., 15, 7307-7336, 10.5194/acp-15-7307-2015, 2015b.

---

## Author Comment (AC1) · 15 Sep 2016

**Response to referee #1: "The effect of particle acidity on secondary organic aerosol formation from α-pinene photooxidation under atmospherically relevant conditions"**

**Yuemei Han et al.**

(*The blue, green, and black fonts represent the referee's comments, the associated revised text in the manuscript, and the authors' responses, respectively.*)

*This work presents laboratory chamber studies on the effect of particle acidity on the secondary organic aerosols (SOA) formation from OH-initiated oxidation of α-pinene, a topic that has received much attention yet still under debate. The authors concentrate their discussions on how seed particle acidity would affect the SOA yield and elemental composition during different stages of photooxidation and under different reaction regimes (high vs. low $NO_x$). Overall, these experimental observations would certainly contribute a relatively complete dataset to studies that highlight the role of aerosol acidity in SOA production and aging.*

Response: We greatly appreciate reviewer #1 for affirming the value of our study and providing thoughtful comments on our manuscript. We have revised the manuscript with careful consideration of all the issues addressed by the reviewer, as described below.

*However, the major conclusions are potentially biased by the experimental protocols, in particular, the interference of $NO_2$ under low $NO_x$ conditions as well as liquid water contents under different particle acidities. The authors are suggested to conduct additional experiments and perform*

*modeling calculations to strengthen their conclusions prior to consideration of being published on*

*Atmospheric Chemistry and Physics.*

Response: We have examined the experimental results carefully and added more detailed discussions regarding the interference of $NO_2$ under low-$NO_x$ conditions and the potential role of liquid water content on the observed acidity effects, as described in the responses to general comments #1 and #3, respectively. We have focused mainly on the further analysis and discussion of the current available experimental data in the revised manuscript, rather than conducting additional experiments proposed by the reviewer. We believe that the proposed experiments will not significantly contribute to the main conclusions of the paper and some of them are likely not appropriate. We have calculated the pH value of seed particles using the Extended Aerosol Inorganic Model (E-AIM) and added more discussions on the relevance of the studied particle acidity to atmospheric conditions, as described in the response to general comment #4.

*General Comments*

*1. Low vs. High $NO_x$ conditions*

*1.1. The authors observed enhanced nitrogen-containing organic fragments formation with increasing aerosol acidity under high $NO_x$ conditions. However, the fraction of these fragments remains constant at different particle acidities under low $NO_x$ conditions, as shown in Figure 6. By examination of the ozone profile shown in Figure 1, one realizes that moderate ozone production is still occurring under the so-called 'low $NO_x$ conditions', owing to the photolysis of $NO_2$ that are either released from the chamber wall or penetrated from the enclosure air, which is OK and cannot be completely avoided in most chamber setups. The production of ozone indicates a rapid recycling of the NO-$NO_2$ chemistry, and thus the active role of NO in the production of the nitrogen-*

*containing organic fragments via the RO$_2$+NO pathway under 'low NO$_x$ conditions'. As shown in Figure 6, the fraction of these nitrogen-containing organic fragments under low NOx conditions seems comparable with those under high NO$_x$ conditions, indicating that they cannot be simply artifacts in the AMS measurement. But the question is why the dependence of these nitrogen-containing organic fragments on particle acidity is only observed under high NO$_x$ conditions? The authors need to give a convincing explanation on this inconsistency.*

Response: We agree that some organic nitrate species might also be formed under low-NO$_x$ conditions. Given that the NO concentration in the chamber was very low (<0.3 ppb) prior to α-pinene photooxidation (i.e., turned on the lamp) under low-NO$_x$ conditions, the contamination of NO$_2$ from the enclosure air was likely negligible. It is most likely that some NO$_2$ was released from the chamber walls and was involved in the α-pinene photooxidation reactions. These organic nitrate species might have different formation pathways and also different composition compared to those formed under high-NO$_x$ conditions, and therefore the acid-catalyzed reactions for these species were possibly insignificant. We have added the following explanations for the observed organic nitrate fragments (i.e., C$_x$H$_y$N$_p^+$ and C$_x$H$_y$O$_z$N$_p^+$) under low-NO$_x$ conditions:

"Noted that a small amount of C$_x$H$_y$N$_p^+$ and C$_x$H$_y$O$_z$N$_p^+$ fragments were also observed under low-NO$_x$ conditions, where NO was not added (Figure 6b). This may be contributed by the formation of minor amounts of organic nitrates from the reactions of NO$_2$ released from the chamber walls with α-pinene oxidation products. The average NO$^+$/NO$_2^+$ ratio was in the range of 6.92–7.91 for particles with different acidities under low-NO$_x$ conditions, which indicates that some organic nitrate species different from those under high-NO$_x$ conditions might be formed. No apparent changes are observed in the mass fractions of C$_x$H$_y$N$_p^+$ and C$_x$H$_y$O$_z$N$_p^+$ fragments with particle acidity under low-NO$_x$

conditions, suggesting that acid-catalyzed formation and partitioning of those organic nitrate species were possibly insignificant." (Lines 368–374)

*1.2. Another issue is that the so-called 'high $NO_x$ condition' does not necessarily lead to the $RO_2$+NO reaction dominant regime over the entire course of the experiment. As shown in Figure 1, NO concentration seems to be completely depleted after a few min of reaction, which is expected due to the intense formation of nitric acid and its rapid deposition on the wall and particles.*

Response: The reviewer could be somewhat correct in terms of "the intense formation of nitric acid and its rapid deposition". However, it is speculative what the nitric acid would do once it deposited on the wall and particles. In order to identify the dominant reaction pathway of organic peroxy radicals ($RO_2$) under high-$NO_x$ conditions, we have assessed the gas-phase reactions in the chamber using a box model based on the Master Chemical Mechanism (MCM, http://mcm.leeds.ac.uk/MCMv3.3.1/home.htt) constrained by the initial experimental conditions in this study. The fraction of $RO_2$ radicals reacted with NO compared to the total reacted $RO_2$ radicals (with NO, $HO_2$, and $RO_2$) was derived from the ratio of $k_{NO}[NO] / (k_{NO}[NO] + k_{HO2}[HO_2] + k_{RO2}[RO_2])$, where $k_{NO}$, $k_{HO2}$, and $k_{RO2}$ represent the reaction rates of $RO_2$+NO, $RO_2$+$HO_2$ and $RO_2$+$RO_2$, respectively and [NO], [$HO_2$], and [$RO_2$] represent the concentration of NO, $HO_2$, and $RO_2$, respectively from the MCM model output. The result from the box model demonstrates that more than 99% of $RO_2$ radicals were reacting with NO in the first 2 hours and at least 62% of $RO_2$ radicals were reacting with NO by the end of the experiments. This suggests that the $RO_2$ + NO was the dominant reaction under the high-$NO_x$ conditions in our study. We have added the following statement in the revised manuscript:

"Approximately 62–99% of $RO_2$ radicals reacted with NO over the entire experimental time (totally 6 hours) under high-$NO_x$ conditions in this study, which was estimated based on the Master Chemical

Mechanism constrained by the initial experimental conditions (S1 and Figure S1 in the Supplement)."

(Lines 192–194)

We have also added the following descriptions regarding the MCM model in the supplement:

**S1   High-NO$_x$ regime assessment**

The Master Chemical Mechanism (MCM v3.3.1, http://mcm.leeds.ac.uk/MCMv3.3.1/home.htt) was incorporated into a box model to assess the NO$_x$ regime for the gas-phase reactions of α-pinene photooxidation under high-NO$_x$ conditions. The box model was constrained with the initial experimental conditions including temperature, pressure, and the concentrations of α-pinene, NO, water vapor, and H$_2$O$_2$ for the individual chamber experiments in this study. The photooxidation reaction of α-pinene was simulated for 6 hours with the box model. The output of the box model was the time series of the concentrations of α-pinene, NO, O$_3$, HO$_2$, and organic peroxy radicals (RO$_2$) (molecule cm$^{-3}$) from each time step with a 1-min resolution. The fraction of RO$_2$ radicals reacted with NO compared to the total reacted RO$_2$ radicals (with NO, HO$_2$, and RO$_2$) was calculated by

$$\frac{k_{NO}[NO]}{k_{NO}[NO] + k_{HO_2}[HO_2] + k_{RO_2}[RO_2]}$$

where $k_{NO}$, $k_{HO2}$, and $k_{RO2}$ are the reaction rates of RO$_2$ + NO, RO$_2$ + HO$_2$ and RO$_2$ + RO$_2$, respectively and [NO], [HO$_2$], and [RO$_2$] are the concentration of NO, HO$_2$, and RO$_2$, respectively. The results from the box model are presented in Figure S1. At the start of the simulations, more than 99% of the RO$_2$ radicals were reacting with NO; while by the end of the experiments (after 6 hours), at least 62% of the RO$_2$ radicals continued to react with NO (Figure S1a). The time series for α-pinene, NO, and O$_3$ from the measurements were reasonably well captured by the box model (Figure S1b, c, and d).

[Figure]

Figure S1. (a) Fraction of $RO_2$ reacted with NO compared to the total reacted $RO_2$ radicals for high-$NO_x$ experiments with ammonium sulfate and acidic seed particles. The measured and the modeled time series of the concentrations of (b) α-pinene, (c) NO, and (d) $O_3$ for the high-$NO_x$ experiment with ammonium sulfate particles ($NH_4/SO_4 = 2$). The variations in time for each species in all experiments with acidic particles under high-$NO_x$ conditions are similar to (b), (c), and (d)."

*1.3. The authors are suggested to carry out continuous NO injection after the initial consumption of NO to truly achieve the 'high $NO_x$ condition'. Alternatively, HONO or $CH_3ONO$ can be used as the OH precursor under high $NO_x$ conditions.*

Response: The reviewer's proposed experiments may in fact achieve the high-$NO_x$ conditions for long periods of time with an additional NO source. However, as described above, the high-$NO_x$ experiments

in our study were also dominated by the reactions of $RO_2$ with NO over the entire experimental course. Therefore, we believe it is not necessary to perform the proposed experiments for this study.

*2. First vs. later generation SOA products*

*2.1. The strong dependence of SOA yield on particle acidities under high NOx conditions was only observed during the initial photooxidation stage, where the total SOA mass concentration is less than 2 ug/m³. As the reaction proceeds, SOA yields are eventually comparable at different particle acidities and the authors explained this phenomenon as the unavailability of free $H^+$ due to thick organic coatings. Note that this argument is established under the assumption that the core of the particles are (semi)-solid and there is no particle-phase diffusion over the entire course of several hours of reaction, which is doubtful considering the moderate RH conditions employed in the experiments. One suspects that if the observed SOA yield dependence at the initial particle growth period on the acidity can be simply attributed to the uncertainties in the AMS measurement when the overall organic loadings are extremely low (< 2 ug/m³).*

Response: We have calculated the detection limits of individual species to estimate the uncertainties of the AMS measurement using data acquired from particle-free periods. The detection limits for organics, sulfate, nitrate, and ammonium were 34, 4, 1, and 5 ng $m^{-3}$, respectively. Given that the measured organic aerosol mass was far above the AMS detection limit, the observed acidity effect in the initial photooxidation stage should not be simply attributed to the uncertainties from the AMS measurement. We have added the following statement in the revised manuscript:

"The detection limits of organics, sulfate, nitrate, and ammonium, defined as 3 times the standard deviations of the mass concentrations of individual species (1-min average) in particle-free air, were 34, 4, 1, and 5 ng $m^{-3}$, respectively." (Lines 136–138)

Furthermore, the core-shell theory assumes that the inorganic core was liquid or solid and the organic coating might be semisolid, instead of "the core of the particles are semisolid" as the reviewer stated. The occurrence of particle-phase diffusion has not been ruled out in our study, although we did not discuss this topic in the original manuscript. The generated α-pinene SOA could be in semisolid or in an amorphous solid state and highly viscous over a range of moderate RH (approximately 29–43% in this study). This could easily slow down the diffusion rate of organic molecules. Considering the large uncertainties in the quantification of SOA viscosity, the mixing time of the organic coating with the inorganic core can vary from seconds to days (Renbaum-Wolff et al., 2013). Therefore, the core-shell assumption is indeed possible. We have added the following explanations in the revised manuscript:

"A possible interpretation for such a decrease in yield is that acidic particles (i.e., the inorganic core) were gradually less accessible with increased organic coating on acidic particles, assuming that the diffusion of organic molecules into the inorganic seeds was considerably slowed. This process was indeed possible at the studied final RH (approximately 29–43%), given that SOA could be in an amorphous solid or semisolid state with high viscosity at low to moderate RH (e.g., ≤ 30%) (Renbaum-Wolff et al., 2013; Virtanen et al., 2010)." (Lines 242–247)

*2.2. The authors are suggested to perform additional calibration experiments:*

*1) Prepare a mixture of pinonic acid (a surrogate for α-pinene photo-oxidation products) and ammonium sulfate salt under different acidities, atomize the mixtures into the chamber under controlled RH and T conditions, and sample the suspended particles by AMS. The authors need to verify if AMS measured organic loadings are identical under different seed acidities even at low organic levels. This calibration experiments can also be used as a means to correct the AMS collection efficiency, if different from the default value used.*

*2) Alternatively, the authors could repeat experiments # 9-12, but adding seed particles after only 30 min of reaction. Would the dependence of SOA yield on the organic loadings still be observed?*

Response: These two proposed experiments are associated with the detection limits of the AMS instrument and potential changes in sensitivity and/or collection efficiency of the organics with particle acidity. As explained above, the measured organic mass concentration was much larger than the organic detection limits of the AMS and thus they are reliable despite being low (note that 2 µg m$^{-3}$ is not too low and it is in fact similar to many ambient atmosphere studies). Regarding the AMS collection efficiency issue, we agree that the CE may change with particle acidity, however it is not a concern in our study. This is mainly because we calculated the generated organic aerosol mass using the following method as given in the manuscript:

"Organic mass concentrations derived from AMS measurement were wall-loss corrected according to the decay of sulfate particles in the chamber, i.e., by multiplying the ratios of the initial sulfate concentrations to the instantaneously measured sulfate concentrations." (Lines 141–143).

Consequently, the real-time measured organic mass was normalized by the sulfate mass and thus the CE has been factored out from this calculation. Therefore, the observed acidity effect is not likely caused by any uncertainties associated with the AMS measurement or the CE issue. As a result, we believe there is no need to perform these two proposed calibration experiments. Additionally, there is no evidence that the ionization efficiency (IE) of organics in the AMS is variable with other factors such as acidity, and a constant IE over the course of days is standard practice in the AMS community.

*3. Aerosol liquid water content*

*3.1. The hygroscopic growth curves for ammonium sulfate seed at different acidities are significantly different. At ~50% RH, diameter changes of $(NH_4)_2SO_4$ and $NH_4HSO_4$ are 0% and 20% at*

*equilibrium, respectively, due to water uptake (Seinfeld and Pandis, 2008). For the experimental conditions in this study, i.e., dry seed particles with different acidities at 50-70% RH, the water content would vary significantly from zero to several tens of percentage in mass, depending the amount of sulfuric acid in the particle phase. As a result, the observed 'acidity effect', if any, should really be the coupled effect of both acidities and water content in aerosols.*

Response: We agree that water content in the particles could increase particle diffusivity and thus affect SOA yield, as the aerosol water may serve as a medium to dissolve gas-phase water-soluble organic species, and/or that the observed acidity effect could be a coupled effect of acidity and water content. In our study, the initial liquid water content estimated by the E-AIM were in the same order of magnitude for individual experiments under high-$NO_x$ conditions, that is, on average 5.2, 6.3, and 10.3 µg m$^{-3}$ for experiments with $NH_4/SO_4$ molar ratios of 1.0, 0.5, and 0.2, respectively. Higher SOA yield seem to correspond with higher particle water content. However, changes in SOA yield were not observed for experiments with similarly varied water content but under low-$NO_x$ conditions (Exp. 5–8; Table 1), despite the measured O/C ratio being higher and presumably associated with more oxidized (and thus soluble) species. We therefore conclude that the particle water content likely played a minor role in the observed increase in SOA yield under high-$NO_x$ conditions, particularly since the O/C ratio and organic fragments under these conditions indicated the potential for a less soluble organic fraction. We have added the following discussions in the revised manuscript:

"In addition to the effect of particle acidity, the α-pinene SOA yield was also possibly influenced by the liquid water content in the particles. The initial water content in the seed particles estimated by the E-AIM was on average 5.2, 6.3, and 10.3 µg m$^{-3}$ for high-$NO_x$ experiments with $NH_4/SO_4$ molar ratios of 1.0, 0.5, and 0.2, respectively. Therefore, more water was present in the particles with higher acidity. The higher particle water content could prompt the partitioning of gas-phase water-soluble organic

species by providing a larger medium for their dissolution and therefore potentially increase the SOA yield (Carlton and Turpin, 2013). However, there was no apparent increase in the SOA yield under low-NO$_x$ conditions, even though seed particles with similarly varied water content were used (Exp. 5–8; Table 1) and despite the fact that products with higher O/C (hence higher solubility) were formed (section 3.3). This suggests that the particle water content likely did not contribute substantially to the observed increase in α-pinene SOA yield with acidity under high-NO$_x$ conditions." (Lines 224–233)

*3.2. Another concern is the different phase states of particles could potentially lead to different bouncing effect on the AMS vaporizer, thus changing the collection efficiency for different seed particles.*

Response: We agree that the phase state of particles being sampled into the AMS may affect the CE. However, as explained in the response to general comment #2, the real-time CE has been factored out in the calculation of organic mass concentration. Therefore, only the initial sulfate mass concentration was relevant to the calculated organic mass concentration. The CE for ammonium sulfate particles at moderate RH (approximately 30–80%) did not vary significantly (Matthew et al., 2008). Therefore, the results of our study would not be affected significantly by the CE issue.

*(Matthew, B. M., Middlebrook, A. M. and Onasch, T. B.: Collection Efficiencies in an Aerodyne Aerosol Mass Spectrometer as a Function of Particle Phase for Laboratory Generated Aerosols, Aerosol Sci. Technol., 42(917683900), 884–898, doi:10.1080/02786820802356797, 2008.)*

*3.3. In view of these uncertainties, the authors are suggested to conduct a series of experiments using hydrated seed particles to begin with.*

Response: Based on the above discussion, the observed acidity effect in our study was not likely caused by the different particle water content in individual experiments or AMS collection efficiency that was factored out in the final calculation of organic mass. Therefore, the proposed experiments are not relevant or warranted.

*4. Particle acidity and atmospheric relevant conditions*

*4.1. In view of the main focus of this study, the seed particle acidities need to be estimated based on the AMS measured inorganic composition and thermodynamic models such as E-AIM (http://www.aim.env.uea.ac.uk/aim/aim.php).*

Response: We have calculated the seed particle acidity, i.e., the pH value of aerosols in the aqueous phase, using the outputs from E-AIM, as summarized in Table 1 below. The mass concentrations of $NH_4^+$ and $SO_4^{2-}$ measured by the AMS have been used as the inputs for the E-AIM. The following explanation has been added to the revised manuscript:

"The initial seed composition in each experiment was predicted using the Extended Aerosol Inorganic Thermodynamic Model (E-AIM) II (http://www.aim.env.uea.ac.uk/aim/aim.php) (Clegg et al., 1998). The concentrations of inorganic sulfate, nitrate, and ammonium derived from the AMS measurement as well as the temperature and RH in the chamber were input parameters. The pH of aerosol particles was calculated by $-\log(\gamma \times [H^+])$ using the model outputs, where $\gamma$ and $[H^+]$ are the activity coefficient of $H^+$ and the molar concentration of dissociated $H^+$ (mol $L^{-1}$) in the aqueous phase, respectively." (Lines 151–155)

**Table 1.  Experimental conditions and SOA yields from OH-initiated photooxidation of α-pinene under high- and low-$NO_x$ conditions.**

| Exp. | NH$_4$/SO$_4$ ratio [a] | Initial seed composition, [b] molality (mole kg$^{-1}$) | Aerosol pH [c] | Temp.[d] (°C) | RH [e] (%) | Seed (µg m$^{-3}$) | NO (ppb) | α-pinene (ppb) | ΔHC (µg m$^{-3}$) | ΔM$_0$ (µg m$^{-3}$) | Yield (%) |
|---|---|---|---|---|---|---|---|---|---|---|---|
| **High-NO$_x$ conditions** | | | | | | | | | | | |
| 1 | 2 | (NH$_4$)$_2$SO$_4$ (no liquid phase) | | 24–31 | 47–29 | 4.4 | 66 | 15.9 | 84.2 | 3.5 | 4.2±0.1 |
| 2 | 1 | H$^+$=3.2, NH$_4^+$=15.3, HSO$_4^-$=11.2, SO$_4^{2-}$=3.7 | −1.31 | 23–30 | 58–34 | 8.4 | 69 | 17.6 | 93.4 | 5.2 | 5.6±0.1 |
| 3 | 0.5 | H$^+$=5.0, NH$_4^+$=7.0, HSO$_4^-$=7.5, SO$_4^{2-}$=2.3 | −1.50 | 24–30 | 61–38 | 6.3 | 68 | 13.6 | 71.8 | 4.7 | 6.6±0.1 |
| 4 | 0.2 | H$^+$=7.2, NH$_4^+$=2.7, HSO$_4^-$=6.2, SO$_4^{2-}$=1.9 | −1.66 | 26–34 | 58–33 | 7.9 | 72 | 17.0 | 89.9 | 6.8 | 7.6±0.2 |
| **Low-NO$_x$ conditions** | | | | | | | | | | | |
| 5 | 2 | (NH$_4$)$_2$SO$_4$ (no liquid phase) | | 25–32 | 67–43 | 12.6 | <0.3 | 19.6 | 96.7 | 34.1 | 35.2±1.1 |
| 6 | 1 | H$^+$=1.9, NH$_4^+$=14.8, HSO$_4^-$=8.5, SO$_4^{2-}$=4.1 | −0.93 | 25–32 | 64–37 | 12.6 | <0.3 | 17.4 | 79.1 | 22.6 | 28.6±1.5 |
| 7 | 0.5 | H$^+$=3.2, NH$_4^+$=10.6, HSO$_4^-$=8.0, SO$_4^{2-}$=2.9 | −1.22 | 26–33 | 64–38 | 11.9 | <0.3 | 19.3 | 92.6 | 33.7 | 36.3±1.5 |
| 8 | 0.2 | H$^+$=5.3, NH$_4^+$=4.1, HSO$_4^-$=5.6, SO$_4^{2-}$=1.9 | −1.35 | 24–33 | 66–36 | 11.4 | <0.3 | 19.5 | 88.6 | 28.4 | 32.0±1.9 |
| **Adding seeds after photooxidation** | | | | | | | | | | | |
| 9 | 2 | (NH$_4$)$_2$SO$_4$ (no liquid phase) | | 23–31 | 57–34 | 9.7 | 82 | 20.4 | | | |
| 10 | 0.5 | H$^+$=5.9, NH$_4^+$=7.0, HSO$_4^-$=8.4, SO$_4^{2-}$=2.3 | −1.72 | 24–31 | 56–33 | 12.2 | 72 | 18.5 | | | |
| 11 | 2 | (NH$_4$)$_2$SO$_4$ (no liquid phase) | | 25–33 | 68–42 | 7.4 | <0.3 | 16.1 | | | |
| 12 | 0.5 | H$^+$=4.9, NH$_4^+$=9.6, HSO$_4^-$=9.3, SO$_4^{2-}$=2.6 | −1.64 | 25–33 | 57–33 | 11.4 | <0.3 | 17.3 | | | |

[a] NH$_4$/SO$_4$ molar ratios of ammonium sulfate/sulfuric acid aqueous solution used for atomizing seed particles. [b] Initial seed composition was estimated using the E-AIM II. [c] Aerosol pH was calculated with the E-AIM output. [d] Initial and final temperature inside the chamber. [e] Initial and final RH inside the chamber.

*4.2. The authors are also suggested to discuss how relevant the acidities used in the chamber experiments with atmospheric conditions.*

Response: We have added the following discussions and one reference regarding the relevance of the studied particle acidity to atmospheric conditions:

 "Despite the initial pH value of aerosol particles investigated in this study (−0.93 to −1.72) being in the higher acidity range relative to that generally observed for ambient aerosols, pH values less than −2.0 have been reported for atmospheric aerosol particles and haze droplets (Herrmann et al., 2015). It is therefore expected that the effect of particle acidity observed in this study is relevant to the ambient atmosphere, especially in regions enriched with acidic aerosols, and possibly during initial particle growth via sulfuric acid." (Lines 399–403)

"Herrmann, H., Schaefer, T., Tilgner, A., Styler, S. A., Weller, C., Teich, M., and Otto, T.: Tropospheric Aqueous-Phase Chemistry: Kinetics, Mechanisms, and Its Coupling to a Changing Gas Phase, Chem. Rev., 115, 4259–4334, doi:10.1021/cr500447k, 2015." (Lines 481–483)

*Minor Comments*

*1. Page 2, Line 50: Ester dimers are observed as an important class of products from the ozonolysis of α-pinene (e.g., Zhang et al., PNAS, 2015). Epoxides are certainly produced, but from a minor pathway (e.g., Eddingsaas et al., ACP, 2012), and ring-opening reactions proceed rapidly following the reactive uptake of epoxides.*

The sentence referred to: "The oxidation of α-pinene by hydroxyl radicals (OH), ozone ($O_3$), and nitrate radicals produces a variety of multifunctional organic compounds such as carboxylic acids, carbonyls, peroxides, epoxides, alcohols, and organic nitrates (Yasmeen et al., 2012)."

Response: We have added "ester dimers" as one of the possible α-pinene oxidation products. We kept "epoxides" despite it being a minor product, because the listed examples here are not necessary intended to be for major products only. The original sentence has been revised to:

"The oxidation of α-pinene by hydroxyl radicals (OH), ozone ($O_3$), and nitrate radicals produces a variety of multifunctional organic compounds such as carboxylic acids, carbonyls, peroxides, ester dimers, epoxides, alcohols, and organic nitrates (Calogirou et al., 1999; Yasmeen et al., 2012; Zhang et al., 2015)." (Lines 52–55)

*2. Page 2, Line 57: It has been well recognized that sulfate esters contribute to a large fraction of SOA mass via reactive uptake of IEPOX, a second generation oxidation products from isoprene photooxidation under low NO conditions, onto acidified aerosols. However, the exact mass fraction*

*of sulfate esters in the α-pinene system has not yet been determined. Surratt et al. (2007) and (2008) report for the first time the evidence of organosulfate in α-pinene derived SOA, but not quantification.*

The sentence referred to: "Enhanced aerosol acidity led to the formation of sulfate esters that contributed to a large fraction of SOA mass from photooxidation of α-pinene under high-$NO_x$ conditions (Surratt et al., 2007a, 2008)."

Response: We agree that the mass fraction of sulfate esters derived from α-pinene photooxidation has not been quantified exactly, although sulfate esters from isoprene photooxidation and in the ambient atmosphere may contribute to a large fraction of SOA mass. As this entire paragraph is intended to quantitatively characterize the acidity effect, we have removed this sentence from the revised manuscript.

*3. Page 5, Line 132: Have the authors performed experiments monitoring the decay of pure sulfate particles? How was the decay rate compared with particles coated with organics?*

The sentence referred to: "Organic mass concentrations derived from AMS measurement were wall-loss corrected according to the decay of sulfate particles in the chamber, i.e., by multiplying the ratios of the initial sulfate concentrations to the instantaneously measured sulfate concentrations."

Response: We did not perform specific experiments monitoring the decay of pure sulfate particles. Particle wall loss rates generally depend on particle size, electrical charge, and turbulence level inside the chamber. With the coating of organics on sulfate particles, the particle size increased less than 100 nm in mobility diameter during the experiments in our reaction system. The decay rate of particles coated with organics could be slightly lower than pure sulfate particles, because larger particles usually

have smaller Brownian diffusion rates. We assumed that particles coated with organics have same deposition rate as pure sulfate particles in this study. This assumption should not contribute significant uncertainty or change the conclusion of the paper. We have added the following statement in the revised manuscript:

"The decay rate of particles coated with organics was assumed to be same as that of pure sulfate particles, although the later could be slightly higher due to the larger Brownian diffusion rate of smaller particles." (Lines 146–148)

**4. Page 5, Line 134: How was the 0.1 ug/cm$^{-3}$ mass from self-nucleation estimated?**

The sentences referred to: "This assumption is appropriate given that organics contributed by self-nucleation was estimated to be less than 0.1 µg cm$^{-3}$ in the studied system."

Response: This estimation is based on the particle number concentration measured by the CPC and the organic mass concentration measured by the AMS, not obtained from the SMPS measured particle size distribution. We observed less than 50 cm$^{-3}$ particles from self-nucleation and also no obvious increase in organic mass concentration by the AMS measurement during the experiments without adding seed particles. We therefore estimated a maximum concentration of 0.1 µg cm$^{-3}$ for self-nucleated organic aerosols by using a particle number concentration of 50 cm$^{-3}$ and a particle diameter of 70 nm. However, this estimation may have some uncertainties in terms of the assumed particle size. We have replaced the estimated value with the observed experimental results:

"This assumption is appropriate given that less than 50 particles cm$^{-3}$ were contributed by self-nucleation and that an obvious increase in organic mass concentration was not observed from the AMS measurement in the experiments without adding seed particles." (Lines 144–146)

*5. Page 5, Line 139: What is the estimated NO₃ photolysis rate?*

The sentences referred to: "The OH concentrations were calculated to be approximately $4.3$–$5.9 \times 10^6$ and $0.8$–$1.1 \times 10^6$ molecules $cm^{-3}$ for experiments under high- and low-NO$_x$ conditions, respectively."

Response: We agree that the photolysis rate of $NO_3$ radicals is an uncertainty in this estimation, since any $NO_3$ present can react with α-pinene and affect subsequent estimations of the OH radical levels. However, the estimated OH concentration was not used for any quantitative analysis in this study. We therefore have added the following explanation regarding this uncertainty in the revised manuscript:

 "Nitrate radical ($NO_3$) generated from the reactions such as $NO_2$ with $O_3$ might also affect the α-pinene decay (and hence the estimated OH), whereas it was not taken into account here because $NO_3$ levels were likely to be small under the studied irradiation conditions." (Lines 159–161)

---

## Author Comment (AC2) · 15 Sep 2016

**Response to referee #2: "The effect of particle acidity on secondary organic aerosol formation from α-pinene photooxidation under atmospherically relevant conditions"**

**Yuemei Han et al.**

(*The blue, green, and black fonts represent the referee's comments, the associated revised text in the manuscript, and the authors' responses, respectively.*)

*The authors performed laboratory chamber experiments to study the effect of particle acidity on α-pinene SOA. Firstly, the authors found that the particle acidity has small effect on α-pinene SOA yield under low-NOx conditions, but has large effects on α-pinene SOA yield under high-NOx conditions. This has been shown in Eddingsaas et al. (2012a). Secondly, the authors showed that α-pinene SOA formation under low-$NO_x$ conditions is influenced if the particle seed is injected after α-pinene photooxidation. This has also been shown in Eddingsaas et al. (2012a). Thirdly, the authors observed that the fraction of $C_xH_yN_z^+$ and $C_xH_yO_zN_p^+$ fragments in total organics increases with particle acidity, which is the only new finding in the manuscript. Considering the lack of novel findings in the manuscript, I would not recommend this manuscript for publication in its current state.*

Response: We thank reviewer #2 for the comments on our manuscript. The reviewer has proposed some relevant points that could improve the quality of this paper. However, it seems that the reviewer has missed the main concept of our study on some points, as will be expanded upon further below. Regardless, the reviewer's conclusion that this study lacks novelty is inaccurate and it is further

inappropriate to make such a conclusion by simply comparing with Eddingsaas et al. (2012a). However, we agree that perhaps we could have improved the description in the paper of how this current study is different from Eddingsaas et al. (2012a), and specifically what aspects are novel.  We assumed that the novelty of the study's results would be apparent, however we admit that additional clarity is warranted. Our study is unique as it investigates the effect of aerosol acidity under conditions more relevant to the ambient atmosphere. The originality and the differences of our paper compared to Eddingsaas et al. (2012a) have been summarized in the response to major comment #1. Clearly there are enough differences and novelty to warrant publication. Moreover, a single study is generally far from enough to demonstrate a scientific fact, especially since some of the current results contradict those of Eddingsaas et al. (2012a). We have addressed all the issues noted by the reviewer and have made the relevant changes in the manuscript.

*Major comments:*

*1.1 This manuscript does not represent a substantial contribution to scientific progress, because most of the findings have been shown in Eddingsaas et al. (2012a). The authors should try to differentiate this study from Eddingsaas et al. (2012a).*

Response: We strongly disagree with this viewpoint, although as noted above we could have made the differences more apparent. Our study does not simply replicate or follow the work of Eddingsaas et al. (2012a). Our study was originally designed to investigate the effect of particle acidity on α-pinene SOA formation under relevant ambient conditions as opposed to Eddingsaas et al. Furthermore, there are a number of other substantial differences between our study and Eddingsaas et al. in terms of studied conditions and research focus. The difference in experimental conditions between the two studies is summarized in the table below. The modest relative humidity and the lower initial concentrations of

seed particles and α-pinene applied in our study are more relevant to the atmosphere compared to Eddingsaas et al. (2012a). In fact, one could argue that the experiments of Eddingsaas et al. (2012a) were conducted under conditions that were entirely irrelevant to the atmosphere, as $NO_x$ levels of 800 ppb are never encountered and RH of <10% are similarly rare.

| | RH (%) | Seed size (nm) | Seed volume ($\mu m^3\ cm^{-3}$) | α-pinene (ppb) | High-$NO_x$ level (ppb) |
|---|---|---|---|---|---|
| Eddingsaas et al. (2012a) | < 10 | 60 | ~10–15 | 19.8–52.4 | 800 |
| This study | 33–67 | 150 | ~2.5–7.1 | 13.6–20.4 | 66–82 |

Regarding the research focus, we investigate the effect of particle acidity at four acidic levels; we observed that the acidity effect was stronger in the initial photooxidation period under high-$NO_x$ conditions; we reported the time scale of the acidity effect, the high-resolution organic fragment distributions, the oxidation state of α-pinene SOA in the different experiments, and the potential formation of organic nitrates based on the AMS measurements. In contrast, there is little or no discussion on these topics by Eddingsaas et al. (2012a). In order to highlight the novelty of our study, an explicit research objective has been added in the revised manuscript:

"The yield of α-pinene SOA was obtained at various particle acidity levels under high- and low-$NO_x$ conditions. The dependence of SOA yield on particle acidity and the time scale of the acidity effect are characterized and discussed. The effect of particle acidity on the chemical composition of α-pinene SOA, the fragment distributions of bulk organics, and the oxidation state of organics are examined based on the high-resolution analysis of organic aerosol mass spectra. The possible contribution of particle acidity to the formation of particulate organic nitrates under high-$NO_x$ conditions is also discussed. Finally, the potential significance of the observed acidity effect in the ambient atmosphere is summarized." (Lines 83–89)

*1.2. For example, while the RH was below 10% in Eddingsaas et al. (2012a), the experiments were conducted under humid conditions in this study. Thus, discussions on dry vs. humid could be added.*

Response: It is true that RH is an important factor in the formation of α-pinene SOA. However, given that we did not conduct experiments under dry conditions, a direct comparison between the dry and humid conditions are not available from this study. We could roughly compare our results with those from previous studies under dry conditions, but other experimental conditions such as temperature and hydrocarbon loading are generally different as well. Therefore, rather than to discuss the dry and humid condition itself, we have added the following statement to indicate the importance of RH in SOA chamber studies:

"Moreover, we have studied the acidity effect under more realistic RH conditions. While RH is an important factor affecting the concentrations of [H$^+$], the kinetics of hydrolysis reactions, and the physical properties of SOA such as viscosity, more investigation over a broader RH range are essential to understand the acidity effect in the real atmosphere." (Lines 403–406)

*1.3. In addition, the authors should provide more insights into why the effects of acidic seed are different between low-NO$_x$ and high-NO$_x$ conditions.*

Response: The acidity effect has been discussed mostly in section 3.2 of the manuscript. The different effects of particle acidity were most likely associated to the distinct reaction mechanisms between high- and low-NO$_x$ conditions, which resulted in the different chemical composition of α-pinene SOA (as seen in Figure 6). It was possible that acid-catalyzed heterogeneous reactions occurred under high-NO$_x$ but not low-NO$_x$ conditions. However, given that the bulk organic aerosol mass was measured by the

AMS (with very strong fragmentation), it is a challenge to provide detailed reaction mechanisms from this study. Therefore, we have proposed this topic to be an open question as follows:

"The potential mechanisms leading to the different acidity effects between high- and low-$NO_x$ conditions warrant further investigation." (Lines 379–380)

*2. The authors observed that the mass fraction of $C_xH_yN_z^+$ and $C_xH_yO_zN_p^+$ in total OA increases with particle acidity under high-$NO_x$ conditions. However, the discussions on this observation are highly speculative.*

*(1) In order to explain this observation, the authors propose "organic nitrates may be formed heterogeneously through a mechanism catalyzed by particle acidity". line 191-192, the authors suggest that particle acidity can facilitate the gas phase RO2 and NOx reaction and organic nitrate formation. This mechanism is highly speculative. The authors need to provide more evidence and cite related reference to support the proposed mechanism.*

The sentence referred to:

a. "The fraction of nitrogen-containing organic fragments ($C_xH_yN_z^+$ and $C_xH_yO_zN_p^+$) in the total organics was enhanced with the increases in particle acidity under high-$NO_x$ conditions, indicating that organic nitrates may be formed heterogeneously through a mechanism catalyzed by particle acidity."

b. "Clearly, $NO_x$ is most likely involved in the acid-catalyzed reactions during α-pinene photooxidation, such as the formation of organic nitrates from $RO_2$ reacting with $NO_x$ facilitated by particle acidity (as discussed in Sect. 3.4)."

Response: The acid-catalyzed formation of organic nitrates could indeed occur, although we agree that is rarely reported in literature. We did not intend to say that acidity can facilitate the $RO_2 + NO_x$ reactions in the gas phase, as the reviewer states. Rather, we are saying that the acidity can enhance the

partitioning into the particle phase and there may also be particle phase reactions which are enhanced by acidity leading to nitrates. As there is no detailed information on the exact form of the organic molecules here, it is not possible to provide a mechanism from the available data of this study. However, one example is the formation of sulfated organic nitrates through the further reactions of sulfuric acid with α-pinene oxidation products of nitroxyl alcohols and carbonyls, as proposed by Surratt et al. (2008). We have revised the original sentences as follows (a and b) and also have added a statement (c) for the possible mechanisms:

a. "…, indicating that organic nitrates may be formed heterogeneously through a mechanism catalyzed by particle acidity *or that acidic conditions facilitate the partitioning of gas phase organic nitrates into particle phase.*" (Lines 23–25)

b. "Clearly, the presence of acidic particles promotes the formation of α-pinene SOA under high-$NO_x$ conditions and $NO_x$ is likely involved in the acid-catalyzed reactions during α-pinene photooxidation." (Lines 217–219)

c. "One possible reaction is the acid-catalyzed formation of sulfated organic nitrates through α-pinene oxidation products such as nitroxyl alcohols and carbonyls reacting with sulfuric acid (Surratt et al., 2008). Further investigations on the individual particle phase organic nitrate species at a molecular level combined with gas-particle kinetics are required to elucidate the detailed reaction mechanisms." (Lines 344–348)

*Surratt, J. D., Gómez-González, Y., Chan, A. W. H., Vermeylen, R., Shahgholi, M., Kleindienst, T. E., Edney, E. O., Offenberg, J. H., Lewandowski, M., Jaoui, M., Maenhaut, W., Claeys, M., Flagan, R. C., and Seinfeld, J. H.: Organosulfate Formation in Biogenic Secondary Organic Aerosol, J. Phys. Chem. A, 112(36), 8345–8378, doi:10.1021/jp802310p, 2008.*

*(2) line 322-323. The change in $NO^+/NO_2^+$ ratio is a reflection of the change in organic nitrate composition, instead of organic nitrate amount. The increasing in $NO^+/NO_2^+$ ratio with particle acidity likely suggests that particle acidity has different effects on the partition of different organic nitrate species. This is one possible explanation for the observation that the mass fraction of organic nitrates increases with particle acidity under high-$NO_x$ conditions.*

The sentence referred: "An increasing $NO^+/NO_2^+$ ratio again suggests that organic nitrates were enhanced with the increase in particle acidity under high-$NO_x$ conditions."

Response: We agree that the composition of organic nitrate species might be different under various acidic conditions and particle acidity could have different effects on the partitioning of different organic nitrate species. The original sentence has been revised to:

"The increase in $NO^+/NO_2^+$ ratio with particle acidity suggests that the composition of organic nitrate species might be different under various acidic conditions, which is possibly due to the varied effect of particle acidity on the formation and/or partitioning of different organic nitrate species." (Lines 364–367)

*(3) As shown in figure 6b, the mass fractions of $C_xH_yN_z^+$ and $C_xH_yO_zN_p^+$ under low-$NO_x$ conditions are similar to that under high-$NO_x$ conditions. How are organic nitrates formed under low-$NO_x$ conditions? Why are the mass fractions of $C_xH_yN_z^+$ and $C_xH_yO_zN_p^+$ under low-$NO_x$ conditions not affected by particle acidity?*

Response: A possible explanation is that a small amount of $NO_2$ released from the chamber walls was involved in the $\alpha$-pinene photooxidation under low-$NO_x$ conditions. The average $NO^+/NO_2^+$ ratio was in the range of 6.92–7.91 for particles with different acidities under low-$NO_x$ conditions, compared to

those of 9.13–10.44 under high-$NO_x$ conditions. This possibly suggests that different organic nitrate species were formed under high- and low-$NO_x$ conditions, and that the particle acidity had different effects on the formation and partitioning of those organic nitrate species. We have added the following statement in the revised manuscript:

"Noted that a small amount of $C_xH_yN_p^+$ and $C_xH_yO_zN_p^+$ fragments were also observed under low-$NO_x$ conditions, where NO was not added (Figure 6b). This may be contributed by the formation of minor amounts of organic nitrates from the reactions of $NO_2$ released from the chamber walls with α-pinene oxidation products. The average $NO^+/NO_2^+$ ratio was in the range of 6.92–7.91 for particles with different acidities under low-$NO_x$ conditions, which indicates that some organic nitrate species different from those under high-$NO_x$ conditions might be formed. No apparent changes are observed in the mass fractions of $C_xH_yN_p^+$ and $C_xH_yO_zN_p^+$ fragments with particle acidity under low-$NO_x$ conditions, suggesting that acid-catalyzed formation and partitioning of those organic nitrate species were possibly insignificant." (368–374)

*(4) line 296. The organic nitrate yield reported in this study is misleading. It is because the fragmentation of organic nitrate in AMS give rise to $C_xH_yO_z^+$, which accounts for a large mass fraction in organic nitrate but not included in the yield calculation in this study. The authors could estimate the organic nitrate yield based on the concentration of the sum of $NO^+ + NO_2^+$ and assumed molecular weight of organic nitrate (A.W.Rollins, 2012; Boyd et al., 2015; Xu et al., 2015a).*

Response: We agree that the original method for the calculation of SOA yield is not appropriate. We have estimated the organic nitrates mass and yield using the method suggested by the reviewer and have added the following details in the revised manuscript:

"Assuming an average molecular weight of organic nitrate molecules ranging from 200 to 300 g mol$^{-1}$, where 62 g mol$^{-1}$ is attributed to the $-ONO_2$ group and the remaining from the organic mass (Boyd et al., 2015), the organic nitrate mass was estimated to be approximately 0.6–1.4 µg m$^{-3}$. This resulted in a contribution of 17.5–20.5% to total α-pinene SOA and an overall organic nitrate yield of 0.7–1.6% under high-NO$_x$ conditions in this study." (Lines 331–335)

*Also, are NO$^+$ and NO$_2$$^+$ included in the SOA yield and O:C calculation?*

Response: As NO$^+$ and NO$_2$$^+$ were considered to be the fragments from organic nitrates in this study, we have included them in the SOA yield and O/C ratio calculation. The mass concentrations of NO$^+$ and NO$_2$$^+$ fragments were generally small and the sum of them was in the range of 0–0.3 µg m$^{-3}$. Therefore, the SOA yield and O/C ratio reported in the revised manuscript only changed very slightly compared to the original values, and these changes did not affect the conclusions in this study.

*3. Many results are confusing.*

*(1) Figure 3: the SOA yield curves under both conditions are not typical and problematic (Griffin et al., 1999). Under low-NO$_x$ conditions, why would SOA yield decrease with delta_Mo at the beginning of the experiments? Under high-NO$_x$ conditions, what causes the SOA yield decrease within the first 30min? If one looks at the figure 1, the yield curve is expected to be monotonic. Is the weird yield curve caused by the SOA wall loss correction? What does the sulfate concentration (measured by AMS) look like over the course of the experiments? Does the sulfate concentration (measured by AMS) change once organics are formed?*

Response: We do not regard the results in Figure 3 as problematic. The main difference in Griffin et al. (1999) compared to our study is that they investigated the SOA yield from ozonolysis of α-pinene without the consideration of an acidity effect. We attribute the gradual decrease in the first 30 minutes to the acidity effect, and more specifically, the decrease in the availability of the acidic phase as organics were formed. This is discussed in the manuscript as follows:

"A slight decrease in the SOA yield for acidic particles was also observed after the relatively higher SOA yields within the first 30 min. A possible interpretation for such a decrease in yield is that acidic particles (i.e., the inorganic core) were gradually less accessible with increased organic coating on acidic particles, assuming that the diffusion of organic molecules into the inorganic seeds was considerably slowed. This process was indeed possible at the studied final RH (approximately 29–43%), given that SOA could be in an amorphous solid or semisolid state with high viscosity at low to moderate RH (e.g., ≤ 30%) (Renbaum-Wolff et al., 2013; Virtanen et al., 2010)." (Lines 241−247)

Also, Figure 3 in our study presents the time series of instantaneous SOA yields vs. $\Delta M_0$ for individual experiments, in contrast to the final SOA yield presented in Griffin et al. (1999). The mass concentration of sulfate decreased smoothly over the experiment period due to particle loss on the chamber wall. There was no significant unexpected change in sulfate concentration once organic mass was formed. The calculated α-pinene SOA yield (i.e., the ratio of $\Delta M_0/\Delta HC$) can be affected by both $\Delta M_0$ and $\Delta HC$. The $\Delta M_0$ in individual experiments increased constantly as seen in Figure 1. The high yields at the very beginning are caused mainly by the very low $\Delta HC$. These data points were acquired by the PTR-MS when α-pinene concentration was high at the very beginning, whereas the calculated $\Delta HC$ was too low and within the α-pinene detection limit. Therefore, we have removed the initial data points with extremely low $\Delta HC$ in Figure 3.

*(2) Figure 4: This figure is confusing. Firstly, following the same delta_Mo, the authors are comparing the SOA yield under different NH₄/SO₄ ratio. The trend shown in figure 4 does not hold if the authors add data points when delta_Mo = ~0.25 ug/m³ (where SOA yield peaks for NH₄/SO₄ = 0.2). Secondly, many data points are obtained from the period when SOA yield decreases with delta_Mo, the reason for which is not clear yet.*

Response: We agree that the data points in the initial period for the experiment with $NH_4/SO_4 = 0.5$ was not in line with the other three experiments ($NH_4/SO_4 = 2$, 1, and 0.2) when making similar plots for $\Delta M_0 = {\sim}0.25$ μg m$^{-3}$ in Figure 4a. However, as these are experimental results, it should be acceptable if any of the experiments did not perfectly match the trend. There could be some uncertainty in the SOA yield for the experiment with $NH_4/SO_4 = 0.5$ at the very beginning. We therefore have used the $\Delta M_0$ values above 0.7 μg m$^{-3}$ for plotting in Figure 4. As SOA yield depended both on $\Delta M_0$ and $\Delta HC$, a decrease of yield indicates a lower aerosol formation potential, which was possibly due to the reduced acidity effect caused by inaccessibility to the acidic medium as organics were formed. That is in fact the point of choosing the first hour to demonstrate that the acidic effect on the yield does indeed change with organic mass increases. We have provided the reason for the decrease of SOA yield in the revised manuscript:

"A possible interpretation for such a decrease in yield is that acidic particles (i.e., the inorganic core) were gradually less accessible with increased organic coating on acidic particles, assuming that the diffusion of organic molecules into the inorganic seeds was considerably slowed. This process was indeed possible at the studied final RH (approximately 29–43%), given that SOA could be in an amorphous solid or semisolid state with high viscosity at low to moderate RH (e.g., $\leq$ 30%) (Renbaum-Wolff et al., 2013; Virtanen et al., 2010)." (Lines 242–247)

*(3) Figure 5: Firstly, the y-axis scale is misleading. Although it seems that SOA yield increases a lot once injecting seed, the actual enhancement is only on the order of 0.01, which is within measurement uncertainty.*

Response: We agree that using "SOA yield" as the label of y-axis is not appropriate here, as this is different from the general definition of SOA yield. The main point of Figure 5 is to show the increase in organic aerosol mass for experiments with acidic particles (high- and low-NO$_x$) compared to those with ammonium sulfate particles. We have changed the "SOA yield" to "Normalized SOA mass" and also added an additional graph in the revised Figure 5 to show the increased organic aerosol mass for the acidic particles compared to the ammonium sulfate particles. The increase in organic aerosol mass was up to 6 times higher for acidic particles than those of ammonium sulfate particles at the initial stages, as presented in Figure 5b below.

[Figure]

Figure 5. (a) The increase of SOA mass with time for experiments with injecting neutral and acidic seed particles after α-pinene photooxidation for 2 and 4 hours under high- and low-NO$_x$ conditions, respectively (Exp. 9–12 in Table 1). Time= 0 hour represents the beginning of reactive uptake of

oxidation products after seed particles added. The SOA mass was normalized by the reacted α-pinene concentration before adding seed particles. (b) The ratio of SOA mass for acidic particles to that of ammonium sulfate particles.

*Secondly, is there organics associated with sulfate seed in the atomizing solution? Based on some tests in our lab, the injection of sulfate would introduce organics, which comes from the atomizing solution, even if HPLC-grade DI water is used to make solution. Actually, the organics associated with sulfate seed may explain the immediate OA increase after adding seed particles (line 239).*

The sentence referred: "Organic aerosol mass increased immediately after adding seed particles for all experiments.

Response: The reviewer is correct in terms of the potential organic contamination during the atomization procedure. We also have noticed that the organic concentration was increased slightly when adding acidic seed particles. However, the organics associated with sulfate seed is not a concern here, as we are discussing the generated organic aerosol mass ($\Delta M_0$), which was obtained by subtracting the initial organic mass from the total organic mass measured in real-time. We have clarified the meaning of the original sentence:

"The **generated** organic aerosol mass increased immediately after adding seed particles for all experiments." (Line 277)

*Thirdly, how is the increase of SOA yields calculated? What's the reference?*

Response: We calculated the SOA yields for the four experiments in Figure 5 using the ratios of $\Delta M_0/\Delta HC$ in the original manuscript. However, as described in the response to the first question, we

believe that it is not appropriate to use "SOA yield" as the label of the y-axis in Figure 5. Therefore, the "SOA yield" has been replaced by the "Normalized SOA mass".

*Fourthly, the results are not consistent with Eddingsaas et al. (2012a) (figure 7), who showed that AS particles have no effect on for both low and high $NO_x$ conditions.*

Response: It is not clear why there was no additional SOA was formed after introducing ammonium sulfate particles in Eddingsaas et al. (2012a). In contrast, the increase of particulate organic aerosol mass in our study can be explained by the reactive uptake of the gas-phase α-pinene oxidation products formed in the early stage by the acidic and ammonium sulfate seed particles. The point here is that the increase of particulate organic aerosol mass was stronger for acidic particles than for ammonium sulfate particles. The results of our study are also not necessary to be consistent with those of Eddingsaas et al. (2012a), considering the different initial conditions. We have added this statement in the revised manuscript:

"This can be explained by the reactive uptake of the gas-phase α-pinene oxidation products formed in the early stages onto the acidic and ammonium sulfate seed particles." (Lines 277–279)

*Minor comments:*

*1. Table 1: (1) Since both $H^+$ and LWC are modeled by E-AIM in the study, I suggest the authors to replace $NH_4/SO_4$ by particle pH, because $NH_4/SO_4$ or ion balance is not a good proxy for particle pH (Guo et al., 2015; Hennigan et al., 2015). (2) Are the [$NH_4$] and [$SO_4$] input for E-AIM obtained from aqueous solution or AMS measurements? The latter should be used, because the $NH_3/NH_4^+$ partitioning would cause the real [$NH_4$]/[$SO_4$] ratio different from that in aqueous solution.*

Response: We have added the aerosol pH value calculated from the E-AIM II in Table 1, as presented below. The mass concentrations of $NH_4^+$ and $SO_4^{2-}$ measured by the AMS have been used as inputs in the E-AIM. The initial seed composition also has been updated accordingly. We still kept the initial $NH_4/SO_4$ molar ratios in Table 1, because they are important references for making ammonium sulfate/sulfuric acid aqueous solution.

**Table 1. Experimental conditions and SOA yields from OH-initiated photooxidation of α-pinene under high- and low-$NO_x$ conditions.**

| Exp. | $NH_4/SO_4$ ratio [a] | Initial seed composition, [b] molality (mole kg$^{-1}$) | Aerosol pH [c] | Temp. [d] (°C) | RH [e] (%) | Seed (µg m$^{-3}$) | NO (ppb) | α-pinene (ppb) | ΔHC (µg m$^{-3}$) | ΔM$_0$ (µg m$^{-3}$) | Yield (%) |
|---|---|---|---|---|---|---|---|---|---|---|---|
| **High-$NO_x$ conditions** | | | | | | | | | | | |
| 1 | 2 | $(NH_4)_2SO_4$ (no liquid phase) | | 24–31 | 47–29 | 4.4 | 66 | 15.9 | 84.2 | 3.5 | 4.2±0.1 |
| 2 | 1 | $H^+$=3.2, $NH_4^+$=15.3, $HSO_4^-$=11.2, $SO_4^{2-}$=3.7 | −1.31 | 23–30 | 58–34 | 8.4 | 69 | 17.6 | 93.4 | 5.2 | 5.6±0.1 |
| 3 | 0.5 | $H^+$=5.0, $NH_4^+$=7.0, $HSO_4^-$=7.5, $SO_4^{2-}$=2.3 | −1.50 | 24–30 | 61–38 | 6.3 | 68 | 13.6 | 71.8 | 4.7 | 6.6±0.1 |
| 4 | 0.2 | $H^+$=7.2, $NH_4^+$=2.7, $HSO_4^-$=6.2, $SO_4^{2-}$=1.9 | −1.66 | 26–34 | 58–33 | 7.9 | 72 | 17.0 | 89.9 | 6.8 | 7.6±0.2 |
| **Low-$NO_x$ conditions** | | | | | | | | | | | |
| 5 | 2 | $(NH_4)_2SO_4$ (no liquid phase) | | 25–32 | 67–43 | 12.6 | <0.3 | 19.6 | 96.7 | 34.1 | 35.2±1.1 |
| 6 | 1 | $H^+$=1.9, $NH_4^+$=14.8, $HSO_4^-$=8.5, $SO_4^{2-}$=4.1 | −0.93 | 25–32 | 64–37 | 12.6 | <0.3 | 17.4 | 79.1 | 22.6 | 28.6±1.5 |
| 7 | 0.5 | $H^+$=3.2, $NH_4^+$=10.6, $HSO_4^-$=8.0, $SO_4^{2-}$=2.9 | −1.22 | 26–33 | 64–38 | 11.9 | <0.3 | 19.3 | 92.6 | 33.7 | 36.3±1.5 |
| 8 | 0.2 | $H^+$=5.3, $NH_4^+$=4.1, $HSO_4^-$=5.6, $SO_4^{2-}$=1.9 | −1.35 | 24–33 | 66–36 | 11.4 | <0.3 | 19.5 | 88.6 | 28.4 | 32.0±1.9 |
| **Adding seeds after photooxidation** | | | | | | | | | | | |
| 9 | 2 | $(NH_4)_2SO_4$ (no liquid phase) | | 23–31 | 57–34 | 9.7 | 82 | 20.4 | | | |
| 10 | 0.5 | $H^+$=5.9, $NH_4^+$=7.0, $HSO_4^-$=8.4, $SO_4^{2-}$=2.3 | −1.72 | 24–31 | 56–33 | 12.2 | 72 | 18.5 | | | |
| 11 | 2 | $(NH_4)_2SO_4$ (no liquid phase) | | 25–33 | 68–42 | 7.4 | <0.3 | 16.1 | | | |
| 12 | 0.5 | $H^+$=4.9, $NH_4^+$=9.6, $HSO_4^-$=9.3, $SO_4^{2-}$=2.6 | −1.64 | 25–33 | 57–33 | 11.4 | <0.3 | 17.3 | | | |

[a] $NH_4/SO_4$ molar ratios of ammonium sulfate/sulfuric acid aqueous solution used for atomizing seed particles. [b] Initial seed composition was estimated using the E-AIM II. [c] Aerosol pH was calculated with the E-AIM output. [d] Initial and final temperature inside the chamber. [e] Initial and final RH inside the chamber.

*2. Figure 1: (1) The legend "OH consumed a-pinene" is confusing because cumulative a-pinene consumed by OH should increase with time instead of decreasing as shown in the figure. I suggest to change the y-axis to "Δa-pinene consumed by OH". (2) Is SOA mass concentration shown in this figure corrected for wall loss?*

Response: (1) As the y-axis is scaled both for total α-pinene concentration and α-pinene consumed by OH, it is not appropriate to change the y-axis to "Δα-pinene consumed by OH". Therefore, in order to avoid confusion, we have changed the "OH consumed α-pinene" to "α-Pinene decay by OH" in the legend of Figure 1.

(2) The SOA mass concentrations were corrected for particle wall loss using the method described in the manuscript. We have added the following statement in the caption of Figure 1: "The presented SOA mass concentrations have been corrected for particle wall loss according to the decay of sulfate mass." (Lines 664–665)

[Figure]

Figure 1. Time series of the mass concentrations of generated SOA and the mixing ratios of NO, $O_3$, total α-pinene decay, and OH consumed α-pinene in (a) high- and (b) low-$NO_x$ experiments using ammonium sulfate as seed particles. Time = 0 hour is defined as α-pinene photooxidation initiated when the lamps were turned on. The presented SOA mass concentrations have been corrected for particle wall loss according to the decay of sulfate particles.

*3. line 15. The core-shell model contradicts with literature. For example, Renbaum-Wolff et al. (2013) measured the viscosity of a-pinene SOA and calculated that the mixing time is on the order of 10-100s under 40-50% RH.*

The sentence referred to: "The SOA yield decreased gradually with the increase in organic mass under high-$NO_x$ conditions, which is likely due to the inaccessibility of the acidity over time with the coating of α-pinene SOA."

Response: The reviewer's comment is not accurate in this case. According to Figure 2 in Renbaum-Wolff et al. (2013), the viscosity of α-pinene SOA varied up to 5 orders of magnitude at 40–50% RH, and the mixing time due to bulk diffusion was in the range of several seconds to $2\times10^4$ seconds (i.e., approximately 5.6 hours). As a result, the proposed core-shell model in this study is reasonable at the studied final RH of approximately 29–43%. Nevertheless, this assumption is based on the upper limit of the estimated viscosity. We have revised the original sentence to:

"The SOA yield decreased gradually with the increase in organic mass **in the initial stage (approximately 0–1 hour)** under high-$NO_x$ conditions, which is likely due to the inaccessibility to the acidity over time with the coating of α-pinene SOA **assuming a slow particle-phase diffusion of organic molecules into the inorganic seeds**." (Lines 14–17)

*4. line 18-20. The authors found that the SOA is more oxidized under low-$NO_x$ conditions than high-$NO_x$ conditions. However, Chhabra et al. (2011) figure 2c showed that the O:C of a-pinene SOA under low-$NO_x$ (using $H_2O_2$) and high-$NO_x$ conditions (using $CH_3ONO$) are similar.*

The sentence referred to: "The fraction of oxygen-containing organic fragments ($C_xH_yO_1^+$ 33–35% and $C_xH_yO_2^+$ 16–17%) in the total organics and the O/C ratio (0.49–0.54) of α-pinene SOA were lower

under high-NO$_x$ conditions than those under low-NO$_x$ conditions (39–40%, 17–19%, and 0.60–0.62), suggesting that α-pinene SOA was less oxygenated in the studied high-NO$_x$ conditions."

Response: The different results between the two studies are most likely due to the distinct experimental conditions. Chhabra et al. (2011) performed their α-pinene photooxidation experiments at lower RHs (4.2% and 4.9%), higher initial α-pinene concentrations (46 and 47 ppb), and generated higher α-pinene SOA loadings (63.9 and 53.7 μg m$^{-3}$), and eventually they obtained lower O/C ratios of 0.40 and 0.42 under low- and high-NO$_x$ conditions, respectively. In contrast, the higher O/C ratios observed in our study (0.52–0.56 and 0.61–0.64 under high and low-NO$_x$ conditions, respectively) suggest that α-pinene SOA was more oxygenated, which is most likely associated to the lower initial α-pinene concentrations used in our reaction system (approximately 14–20 ppb). This is consistent with the following statement in our manuscript:

"Laboratory studies were usually performed with relatively high loadings of hydrocarbons (e.g., from tens of ppb to several ppm), which would result in higher yield and lower oxidation state of laboratory SOA compared to ambient SOA (Ng et al., 2010; Odum Jay et al., 1996; Pfaffenberger et al., 2013; Shilling et al., 2009)." (Lines 70–73)

*5. line 35. It should be "enhance the reactive uptake of gas phase organics", instead of "particle phase".*

The sentence referred to: "The presence of acidic aerosol particles has been shown to enhance the reactive uptake of particle phase organics and increase SOA yields due to acid-catalyzed reactions (i.e., Garland et al., 2006; Jang et al., 2004; Liggio and Li, 2006; Northcross and Jang, 2007)."

Response: The "particle phase organics" has been changed to "gas phase organic species" in the revised manuscript.

**6. line 37. Cite Surratt et al. (2010) and Xu et al. (2015a), who showed the effect of sulfate on the reactive uptake of IEPOX.**

Response: The two references have been cited as below and also listed in the References.

"The presence of acidic aerosol particles has been shown to enhance the reactive uptake of gas phase organic species and increase SOA yields due to acid-catalyzed reactions (i.e., Garland et al., 2006; Jang et al., 2004; Liggio and Li, 2006; Northcross and Jang, 2007; Surratt et al., 2010; Xu et al., 2015a)." (Lines 37–39)

**7. line 43. It is not accurate to state that atmospheric chemistry models do not consider the dependence of SOA formation on aerosol acidity. Large efforts have been devoted to consider the effect of particle acidity for SOA through IEPOX uptake (Marais et al., 2016; McNeill et al., 2012; Pye et al., 2013). Please rephrase this sentence.**

The sentence referred to: "The dependence of SOA formation on aerosol acidity has not been considered in most atmospheric chemistry models thus far due to the large uncertainties associated with its quantification."

Response: The original sentence has been reworded to:

"The dependence of SOA formation on aerosol acidity generally has not been incorporated in many atmospheric chemistry models thus far due to the large uncertainties associated with the quantification

of acidity effects, with the exception of the acidity effect for SOA via isoprene epoxydiol uptake (Marais et al., 2016; Pye et al., 2013)." (Lines 45–48)

*8. line 50-52 and line 63-66. I don't agree with the authors that "large discrepancies among experiments remain with respect to the effects of aerosol acidity on SOA formation". The seemingly "contradictory" observations among studies listed in the manuscript are just due to the difference in experimental conditions. For example, the effects of particle acidity on α-pinene SOA formation are different for low-$NO_x$ and high-$NO_x$ conditions. Thus, the previous studies cited in the manuscript only show the complexity of this scientific question, instead of the discrepancy.*

The sentence referred to:

a. "Despite the efforts of previous laboratory studies, large discrepancies among experiments remain with respect to the effect of aerosol acidity on SOA formation from individual hydrocarbons."

b. "The large discrepancy regarding the previously reported acidity effect on α-pinene SOA is most likely attributed to the varied experimental parameters such as particle acidity, initial hydrocarbon concentration, oxidant type and level, $NO_x$ level, temperature, and relative humidity (RH)."

Response: The "discrepancies" in the above two sentences are actually more close to "dissimilarities", instead of "contradictory". We agree that different results in previous studies show the complexity of this scientific question. To clarify these points, we have revised the original sentences to:

a. "Despite the efforts of previous laboratory studies under various experimental conditions, the effect of aerosol acidity on SOA formation from individual hydrocarbons remains unclear due to the complexity of this scientific question." (Lines 55–57)

b. "These inconsistent results reported previously are most likely attributed to the varied experimental parameters such as particle acidity, initial hydrocarbon concentration, oxidant type and level, $NO_x$ level, temperature, and relative humidity (RH)." (Lines 67–69)

**9. line 124. The collection efficiency of AMS. Was a dryer deployed upstream of AMS and SMPS? If not, considering the high RH in this study, the particle water could affect the comparison between AMS and SMPS.**

The sentence referred to: "A collection efficiency value of 0.7 was applied for the AMS data analysis based upon the comparison of the volume concentrations derived from AMS and SMPS measurements, assuming that particles are spherical and the densities of organics, sulfate, ammonium, and nitrate are 1.4, 1.77, 1.77, and 1.725 g cm$^{-3}$, respectively."

Response: We did not use a dryer upstream of AMS and SMPS during the experiments. As a result, the particle volume concentrations derived from the SMPS could have a contribution by aerosol water, whereas those derived from the AMS does not include water. This is an uncertainty for the collection efficiency estimation. We have added the following statement to demonstrate this uncertainty in the revised manuscript:

"Note that aerosol particles were not dried upstream of the AMS and SMPS measurements, and thus particle water content might have contributed to the SMPS-derived volume concentrations. This was not taken into account for the AMS-derived volume concentration." (Lines 134–136)

**10. line 133-134. How do the authors estimate the concentration of organics from self-nucleation? Is the particle size distribution bimodal? Please show the particle size distribution measured by SMPS.**

The sentence referred to: "This assumption is appropriate given that organics contributed by self-nucleation was estimated to be less than 0.1 µg cm$^{-3}$ in the studied system."

Response: This is same as minor comment #4 from reviewer #1. This estimation is based on the particle number concentration measured by the CPC and the organic mass concentration measured by the AMS, not obtained from the particle size distribution measured by SMPS. We observed less than 50 cm$^{-3}$ particles contributed by self-nucleation and also no obvious increase in organic mass concentration by the AMS measurement in the experiments without adding seed particles. We therefore estimated a maximum concentration of 0.1 µg cm$^{-3}$ for self-nucleated organic aerosols by using a particle number concentration of 50 cm$^{-3}$ and a particle diameter of 70 nm. However, this estimation may have some uncertainties in terms of the particle size. We have therefore replaced the estimated value with the observed experimental results as follows:

"This assumption is appropriate given that less than 50 particles cm$^{-3}$ were contributed by self-nucleation and that an obvious increase in organic mass concentration was not observed from the AMS measurement in the experiments without adding seed particles." (Lines 144–146)

*11. line 166-167. The lower SOA yield under high-NO$_x$ conditions is due to RO$_2$ reacts with NO and likely undergo fragmentation to produce volatile species, not due to the formation of organic nitrates. According to group contribution method by Pankow and Asher (2008), the reduction in vapor pressure by adding of one nitrate functional group is similar to that of adding one hydroxyl group.*

The sentence referred to: "RO$_2$ reacted primarily with NO and NO$_2$ under high-NO$_x$ conditions, which would result in the formation of relatively volatile species such as organic nitrates and reduce the overall SOA yield; …"

Response: We agree that using organic nitrates as an example here is inappropriate, as some organic nitrates are less- or non-volatile. The original sentence has been reworded to:

"$RO_2$ reacted primarily with NO under high-$NO_x$ conditions and generated relatively more volatile products that reduced the overall SOA yield, …" (Lines 189–190)

*12. line 246-247. Eddingsaas et al. (2012a) is not properly cited. Section 3.3 in Eddingsaas et al. (2012a) stated that "Under high-NO2 conditions, no additional SOA is formed after the addition of either neutral or acidic seed particles in the dark". Thus, the finding in this study is not consistent with Eddingsaas et al. (2012a).*

The sentence referred to: "Eddingsaas et al. (2012) similarly reported that α-pinene photooxidation products preferentially partition to highly acidic aerosols when introducing seed particles after OH oxidation under both low-$NO_x$ and high-$NO_2$ conditions."

Response: We agree that this effect was not observed under high-$NO_2$ conditions by Eddingsaas et al. (2012a). The original sentence has been rephrased to:

"Eddingsaas et al. (2012) also reported that α-pinene photooxidation products preferentially partition to highly acidic aerosols when introducing seed particles after OH oxidation under low-$NO_x$ conditions." (Lines 285–287)

*13. line 249-250. The authors propose that the early generation products don't participate in acid catalysis. This is not consistent with Eddingsaas et al. (2012a) (figure 8), who showed that the first generation products can partition to acidic particles.*

The sentence referred to: "The results of Figure 5 also indicate that early α-pinene oxidation products under low-$NO_x$ condition did not participate in acid catalysis, whereas the later products did."

Response: This conclusion is not in conflict with Eddingsaas et al. (2012a). Indeed, the increase of organic mass was only observed when introduced seed particles after photooxidation under low-NO$_x$ conditions. We therefore concluded that early α-pinene oxidation products under low-NO$_x$ condition did not participate in acid catalysis. Note that here the "early α-pinene oxidation products" in our study are not necessary the "first generation products". Figure 8 in Eddingsaas et al. (2012a) presents the time series of CIMS traces in experiments with adding seed particles after 4-hour α-pinene photooxidation under low-NO$_x$ condition, where the products are similar to "the later products" defined in our study. We have rephrased the original sentence to make it more clear:

"The results in Figure 5 also indicate that later products of α-pinene oxidation were more likely to be acid-catalyzed than the early products under low-NO$_x$ conditions." (Lines 287–288)

***14. line 267-270. The authors need to cite previous studies which discussed the gas phase products from a-pinene oxidation. Especially, Eddingsaas et al. (2012b) showed that pinonaldehyde is important intermediate under both low and high-NO$_x$ conditions.***

The sentence referred to: "The dependence of chemical composition and oxidation state of α-pinene SOA on NO$_x$ level is most likely associated with the gas-phase chemistry of RO$_2$. For instance, peroxynitrates and organic nitrates formed from the chemical reaction of RO$_2$ and NO$_x$ are the dominant products under high-NO$_x$ conditions, whereas organic peroxides and acids formed from RO$_2$ with HO$_2$ are dominant under low-NO$_x$ conditions (Xia et al., 2008)."

Response: We intended to address the different final oxidation products from the gas-phase chemistry of RO$_2$ rather than to refer to the detailed gas-phase chemistry here. Therefore, we have rephrased the first sentence slightly as follows:

"The dependence of chemical composition and oxidation state of α-pinene SOA on $NO_x$ level is most likely associated with the different oxidation products from gas-phase chemistry of $RO_2$." (Lines 305–306)

*15. line 270-272. Xu et al. (2014) is not properly cited. Instead of showing that isoprene SOA is more oxidized under low-$NO_x$ conditions, Xu et al. (2014) showed that the oxidation state of isoprene SOA shows a non-linear dependence on $NO_x$ level.*

The sentence referred to: "SOA formed from photooxidation of isoprene has also been reported to become more oxidized under low-$NO_x$ conditions, which is contrary to those under high-$NO_x$ conditions (Xu et al., 2014)."

Response: As stated by Xu et al. (2014), "SOA becomes less volatile and more oxidized as oxidation progresses in $HO_2$-dominant experiments, while the volatility of SOA in mixed experiments does not change substantially over time". Therefore, this sentence itself is correct, but it was not appropriate to be cited here and also it distracted our major focus on α-pinene. We have removed this sentence in the manuscript.

*16. line 318. Please cite Boyd et al. (2015) and Xu et al. (2015b).*

Response: These two references have been cited as follows and also listed in the section of References:
"Large relative contribution of organic nitrates to the nominal inorganic nitrate fragments is demonstrated by a higher $NO^+/NO_2^+$ ratio than those of pure ammonium nitrate (Bae et al., 2007; Boyd et al., 2015; Farmer et al., 2010; Fry et al., 2009; Xu et al., 2015b)." (Lines 359–361)

*17. line 331. What's the Org/SO₄ ratio in this study? Is it atmospherically relevant?*

The sentence referred to: "Given that the α-pinene loading used in this study was low, similar phenomenon may also occur in the atmosphere."

Response: The final ratio of organics to sulfate at the end of each experiment was in the range of 0.6–0.8 and 1.7–2.8 under high- and low-$NO_x$ conditions, respectively. These are relevant to atmospheric levels. We have added this information to the original sentence as follows:

"Given that the α-pinene loading used in this study was low **and the generated organic aerosol mass was relevant to ambient levels (e.g., the final ratio of organic/sulfate was 0.6–0.8 and 1.7–2.8 under high- and low-$NO_x$ conditions, respectively)**, similar process may also occur in the atmosphere." (Lines 382–385)

*18. line 342. The authors have already ruled out the mechanism "acidic seed facilitates the partitioning of gas phase organic nitrate" (line 308-310). Please rephrase this sentence.*

The sentence referred to:

a. "This is inconsistent with the observed increase in organic nitrate fragments with increasing acidity. Therefore, acid-catalyzed formation of organic nitrates also very likely contributed to the observed enhancement of N-containing organic fragments with particle acidity."

b. "Organic nitrates in these experiments may be formed heterogeneously through a mechanism catalyzed by particle acidity and/or the acidic conditions facilitate the partitioning of gas phase nitrates into the particle phase under high-$NO_x$ conditions."

Response: We did not rule out the partitioning mechanism from the original line 308–310. We meant that the partitioning mechanism cannot solely explain the enhanced N-containing organic fragments. We have added the following explanations to clarify the meaning:

"Moreover, it has been demonstrated that acid-catalyzed hydrolysis is an important removal process for organic nitrates in the particle phase, from which organic nitrates can be converted to alcohols and nitric acid (Day et al., 2010; Hu et al., 2011; Liu et al., 2012; Rindelaub et al., 2015). This process would also enhance the partitioning of gaseous organic nitrates into the particle phase due to the perturbation in gas/particle partitioning, and therefore decrease the organic nitrate yields both in the gas and particle phases (Rindelaub et al., 2015). The observed increase in N-containing organic fragments with particle acidity under high-$NO_x$ conditions suggests that the production of organic nitrates generally exceeded their removal rates in this reaction system." (Lines 348–354)

**19. line 495. The author list of this citation is wrong.**

Response: The author list of this citation was correct. Please find the entire author lists from the original line 494.

"Shilling, J. E., Chen, Q., King, S. M., Rosenoern, T., Kroll, J. H., Worsnop, D. R., DeCarlo, P. F., Aiken, a. C., Sueper, D., Jimenez, J. L. and Martin, S. T.: Loading-dependent elemental composition of α-pinene SOA particles, Atmos. Chem. Phys., 9, 771–782, doi:10.5194/acp-9-771-2009, 2009." (Lines 583–585)

---

## Author Response (AR2)

Yuemei Han
Air Quality Research Division
Environment and Climate Change Canada
Dufferin Str., Toronto, Ontario M3H 5T4, Canada
Phone: 647-895-5690
Email: yuemeihan@hotmail.com

October 19, 2016

Dear Prof. Nizkorodov:

We are submitting our revised manuscript (acp-2016-301-version4) for publication in Atmospheric Chemistry and Physics. We have revised the manuscript with consideration of all the comments from reviewer #3. Please see our response to the reviewer below for details. We also have attached a copy of the manuscript with all the changes tracked for other minor corrections. We hope that this paper will be accepted in your journal soon. Thank you very much for editing the paper.

Sincerely yours,
Yuemei Han and coauthors

**Response to referee #3: "The effect of particle acidity on secondary organic aerosol formation from α-pinene photooxidation under atmospherically relevant conditions"**

**Yuemei Han et al.**

*(The blue, green, and black fonts represent the referee's comments, the associated revised text in the manuscript, and the authors' responses, respectively.)*

*This paper explores the effect of aerosol acidity on alpha-pinene SOA formation under low- and high-NOx conditions. Although this system has been examined by prior studies, only a few studies (research groups) have really investigated this question but under varying reaction conditions (i.e., oxidant type, mixing ratios of alpha-pinene, NO, NO2, and NOx, as well as inorganic seed aerosol composition and RH). This has made the interpretation of the effect of acidity on alpha-pinene unclear. As the authors rightly point out in their reply to one of the initial reviewer comments, just because this topic has been examined before doesn't mean the scientific question under study is fully addressed. I think the most important results from this study include, (1) a stronger effect of aerosol acidity on alpha-pinene SOA formation under high-NOx conditions, especially early in the reaction (first hour), compared to low-NOx conditions. The authors suggest that since the SOA yields are larger under low-NOx conditions (even with less acidic aerosol present), it is possible that organic coatings form on the inorganic core and thus prevent later generation products from interacting with the acidic media to undergo acid-catalyzed particle-phase reactions; (2) Under the low-NOx condition, the authors did find if they initially conducted the photooxidation without seed aerosol they didn't see significant nucleation of SOA. This is important as they waited a few hours later to*

*inject inorganic seed aerosol and found the more acidic inorganic seed aerosol yielded more SOA, suggesting that later-generation products produced under low-NOx conditions could in fact reactively uptake onto these acidic particles. To me, this is one of the most interesting and significant findings of these studies conducted here. I think the authors may want to highlight this effect in the abstract! One could imagine the potential implications of this. For example, if NOx conditions are low enough and alpha-pinene produces later-generation products from a forested area that are then transported near a power plant plume or urban area, you might have acid enhancement of alpha-pinene SOA as a result; and (3) I think the acid enhancement of organic nitrates in the aerosol phase in the high-NOx experiments is quite interesting and potentially important! For example, Surratt et al. (JPCA, 2008) showed the the presence of acidic sulfate aerosol yielded significant quantities of nitrated organosulfates. It is possible that high-volatility organic nitrate products are converted into lower volatility products (like organosulfates), especially if the nitrate group is on a favorable carbon type (i.e., primary, secondary, or tertiary). Professor Matthew Elrod's group at Oberlin College has published work on how stable different organic nitrates are stable in acidic aqueous media. These studies might be helpful in supporting your results or at least provide insights into which types of organic nitrates remain unhydrolyzed in the SOA particles. Overall, I think this is a well-written paper that warrants publication in ACP after the authors consider my specific comments below. I think the authors did a great job citing all of the relevant published work.*

Response: We greatly appreciate the reviewer for the insightful comments in regards to our study. We have carefully considered all the issues addressed by the reviewer and revised the manuscript accordingly, as described below.

Regarding the acidity effect on the later-generation SOA, we have highlighted it by adding the following statement in the Abstract and Implications sections of the revised manuscript:

"This effect could be important in the atmosphere under conditions where α-pinene oxidation products in the gas-phase originating in forested areas (with low $NO_x$ and $SO_x$) are transported to regions abundant in acidic aerosols such as power plant plumes or urban regions." (Lines 19–21 and 407–410)

We have addressed the possible acid-catalyzed formation of organic nitrates as proposed by Surratt et al. (JPCA, 2008) in the manuscript as follows:

"One possible reaction is the acid-catalyzed formation of sulfated organic nitrates through α-pinene oxidation products such as nitroxyl alcohols and carbonyls reacting with sulfuric acid (Surratt et al., 2008)." (Lines 356–358)

In addition, Bleier and Elrod (JPCA, 2013) reported that there were no long-lived organic nitrate species observed from aqueous phase reactions of α-pinene oxides, suggesting the quick hydrolysis of organic nitrate species in bulk aqueous solutions. However, given that the chamber reaction system in our study is quite different than a bulk solution, we did not include the discussions regarding the stability of the organic nitrates in our study compared to that of Bleier and Elrod (JPCA, 2013).

*Specific Comments to Consider Before Publication:*

*1.) There is concern that $H_2O_2$ used in low-NOx chamber experiments may bias a larger production of organic hydroxyhydroperoxides than what might actually occur in the atmosphere. This has been recently discussed in recent results published on isoprene SOA by Liu et al. (2016, ES&T). The reason this matters is this may make SOA yields much higher than expected in the atmosphere since many of these hydroperoxides are multifunctional and are ELVOC like. Thus, you may make a lot of SOA in chamber experiments due to the high HO2 levels but maybe not a lot in the atmosphere through these compounds. This is not answered question, but I think the authors may want to put some word of caution on the use of H2O2 in chamber studies.*

Response: We agree that the $H_2O_2$ level is possibly an important factor affecting the SOA yield derived from laboratory experiments. The following statement has been added in the revised manuscript:

"The oxidant used in laboratory studies is also possibly one of the important factors affecting SOA formation. For example, a positive dependence of SOA yield on $H_2O_2$ level has been reported for the photooxidation of isoprene (Liu et al., 2016)." (Lines 77–79)

*2.) Coatings - I agree that organic coatings could be a major reason for the observed effects you reported. I noticed that the authors probably didn't cite a recent paper that wasn't published at the time of this submission that demonstrated that a-pinene SOA coatings derived from a-pinene ozonolysis suppressed the reactive uptake of isoprene epoxydiols (IEPOX) (Riva et al, 2016, ES&T). Although only a minor point in this prior study, Lin et al. (2014, ES&T) did demonstrate that acid-catalyzed reactive uptake of IEPOX could be self-limiting depending on the initial inorganic seed aerosol. Specifically, when they injected the same amount of IEPOX into a chamber filled with either MgSO4 + H2SO4 or (NH4)2SO4 + H2SO4 they found that the SOA yield was MUCH MUCH lower for the former case due to the production of light-absorbing oligomers. It appeared that these oligomers may have formed a diffusion barrier for further uptake of IEPOX. Furthermore, Professor Faye McNeill's group at Columbia University demonstrated that the reactive uptake of alpha-pinene oxide is also self-limiting (Drozd et al., 2014, ACP). The point of me in raising these studies is I think it provides more credibility for the interpretation of your results.*

Response: We thank the reviewer for raising this good point in support of the coating theory in this study. We have added the following statements regarding the organic coating and the self-limiting of their reactive uptake in the revised manuscript:

"A possible interpretation for such a decrease in yield is that acidic particles (i.e., the inorganic core) were gradually less accessible with increased organic coating on acidic particles. **This assumes that a phase separation of particulate organic and inorganic components occurred, from which a core-shell morphology is inferred (Drozd et al., 2013), and that** the diffusion of organic molecules into the inorganic core was considerably slowed." (Lines 248–252)

"It is expected that further reactive uptake of α-pinene SOA to acidic particles might have been suppressed due to a phase separation, as has been reported by other studies (Drozd et al., 2013; Lin et al., 2014; Riva et al., 2016)." (Lines 255–257)

*3.) Vapor losses to Chamber - The time scale of your chamber experiments is quite long, which could be a problem for low vapor pressure products in a small chamber like yours. How might wall losses of ELVOC-like species (Ehn et al., 2014, Nature) affect the SOA yields you report here? It is becoming more common practice now to use CIMS instruments to measure wall losses of gaseous species in order to provide more accurate estimates of SOA yields (Zhang et al., PNAS, 2015). The question I'm posing to the authors is since you focus on reporting SOA yields, couldn't the yields you report actually be an underestimate due to wall loss issues related to "sticky" vapors? I think this has to be acknowledged in the text.*

Response: We agree that the SOA yield reported in our study might have been somewhat underestimated due to vapor wall loss. This has been addressed in the manuscript:

"The calculated SOA yield could have been affected by the wall loss of semi-volatile vapors at low α-pinene loadings, whereas this effect was not taken into account herein."

Also, in the high $NO_x$ case we expect that the formation of ELVOC will be highly diminished, since $RO_2$ + NO reactions will dominate over auto-oxidation. Nevertheless, we have reworded this sentence slightly to highlight the vapor wall loss effect as follows:

"The calculated SOA yield could have been affected by the wall loss of **vapors at low α-pinene loadings, in particular for low- and semi-volatile gaseous species (Ehn et al., 2014); however, such an** effect was not taken into account herein." (Lines 154–156)

*4.) Something remains unclear to me. In the experiments using pure ammonium sulfate seed aerosol, do the authors think these particle effloresced? If so, how can you accurately calculate the acidities of these particles? Further, if they did efflorescence, do you think the lack of aerosol water on these particles could have affected the potential multiphase chemistry?*

Response: We did not report the particle acidity for the experiments using pure ammonium sulfate seed particles (as seen in Table 1 of the manuscript), since there was no aqueous phase based on the estimation with the E-AIM model. Given that the efflorescence relative humidity for ammonium sulfate particles with a dry diameter of 150 nm is approximately 30% (e.g., Gao et al., 2006; Saukko et al., 2015), the efflorescence of these particle possibly occurred to a degree at the studied RH. Previous studies have demonstrated that α-pinene SOA yield increased with the increase in RH (e.g., Jonsson et al., 2006; Kristensen et al., 2014; Zhang et al., 2015). The lack of aerosol water could increase the timescale of particle-phase diffusion and the mean molecular weight of the particulate organic species, and also possibly affect the reactive uptake of certain compounds. This is an open question which warrants further investigations. However, the particle water content likely did not contribute substantially to the observed increase in α-pinene SOA yield with acidity in this study, as discussed in section 3.2.1.

*Gao, Y., Chen, S. B., and Yu, L. E.: Efflorescence relative humidity for ammonium sulfate particles, J. Phys. Chem. A, 110(24), 7602–7608, doi:10.1021/jp057574g, 2006.*

*Saukko, E., Zorn, S., Kuwata, M., Keskinen, J., and Virtanen, A.: Phase state and deliquescence hysteresis of ammonium-sulfate-seeded secondary organic aerosol, Aerosol Sci. Technol., 6826(June 2015), 00–00, doi:10.1080/02786826.2015.1050085, 2015.*

*Zhang, X., McVay, R. C., Huang, D. D., Dalleska, N. F., Aumont, B., Flagan, R. C., and Seinfeld, J. H.: Formation and evolution of molecular products in α-pinene secondary organic aerosol., Proc. Natl. Acad. Sci. U. S. A., 112, 14168–73, doi:10.1073/pnas.1517742112, 2015.*

**5.) Lack of acidity effect under low-NOx conditions: The lack of an acidity effect is interesting! Recent work by Liu et al. (2015, PCCP) demonstrated the hydroperoxides might react on acidic particles and off gas more volatile products? Do you think this could be happening here? One way to check is to examine your data where you conduct the photooxidaiton experiments without seed aerosol but later add in the acidic aerosol. Does your PTR-MS reveal enhanced volatile product formation after this introduction of seed aerosol under low-NOx conditions? Further, I wonder if you see changes in the aerosol composition with the AMS? That might reveal something about potential multiphase chemistry? It's possible with the AMS you see very different OA mass spectra when you add in the seed aerosol, suggesting something is occuring. This kind of connection to the chemical data you have could provide more insights into the SOA yields you report.**

Response: The dependence of reactive uptake on neutralization (i.e., aerosol acidity) varies by species, as reported by Liu et al. (2016, PCCP). It is possible that certain organic compounds such as hydroperoxides might react on acidic particles and produce volatile products. This would be a plausible explanation for the different acidity effects between high- and low-$NO_x$ conditions observed here.

Further studies are warranted to elucidate the gas-phase chemistry in this reaction system. It is a challenge to provide a detailed characterization of the gas-phase oxidation products here, as there is no quantitative measurement of those species by the PTR-MS in this study, and the normalized PTR-MS spectra were not significantly different regardless. However, we have compared the mass spectra of α-pinene SOA between the experiments with and without acidic seed particles under low-$NO_x$ conditions. As seen from the figure below, the mass spectra from the two conditions are quite different, and it seems that the α-pinene SOA formed by adding acidic seed particles after photooxidation are more oxygenated than those of adding acidic seed particles before photooxidation. This could suggest that volatile organic species are possibly produced in the presence of acidic particles under low-$NO_x$ conditions. Therefore, we have added the following statement in the revised manuscript:

"In addition, some oxidation products such as hydroperoxides might have reacted on the acidic particles and produced more volatile products (Liu et al., 2016), which may manifest as a decrease in the acidity effect (i.e., lower yield) for α-pinene SOA under low-$NO_x$ conditions." (Lines 334–336)

Liu, Y. J., Kuwata, M., McKinney, K. A., and Martin, S. T.: Uptake and release of gaseous species accompanying the reactions of isoprene photo-oxidation products with sulfate particles, Phys. Chem. Chem. Phys., 18(3), 1595–1600, doi:10.1039/C5CP04551G, 2016. (Lines 570–572)

[Figure]

(This Figure presents the high-resolution mass spectra of α-pinene SOA for low-NO$_x$ experiments with adding acidic seed particles (molar ratio of NH$_4$/SO$_4$ = 0.5) (a, Exp. 12 in Table 1) after and (b, Exp. 7 in Table 1) before the photooxidation. The mass spectra were averaged in the first two hours of the reactive uptake of oxidation products after seed particles were added.)

*6.) The authors say the particles were coated with SOA. How is this known? Are you assuming this based on prior studies that related phase separation to O:C ratios (which is fine, but wanting to make sure I'm clear on how you know it is a coating versus mixture). Or did you conduct microscopy measurements to know this?*

Response: The coating theory in our study is an assumption based on previous studies and we did not conduct aerosol microscopy measurements. This was partly explained in the previous version. We have provided more explanations to make this point clearer in the revised manuscript as follows:

"A possible interpretation for such a decrease in yield is that acidic particles (i.e., the inorganic core) were gradually less accessible with increased organic coating on acidic particles. **This assumes that a phase separation of particulate organic and inorganic components occurred, from which a core-shell morphology is inferred (Drozd et al., 2013), and that** the diffusion of organic molecules into the inorganic core was considerably slowed." (Lines 248–252)

*7.) Is it possible to show average mass spectra from each condition in the SI? I'm curious to know if the authors find any marker ions at higher m/zs that could potentially be used as tracers? For example, Lin et al. (2012, ES&T) and Budisulistiorini et al. (2013, ES&T) showed that m/z 82 could be a direct marker ion for the acid-catalyzed reactive uptake of IEPOX. This has been later confirmed by Hu et al. (2015, ACP).*

Response: The average mass spectra of α-pinene SOA under different experimental conditions (see below) have been presented in Figure S2 of the supplement. We did not find strong evidence of specific tracer ions related to acid-catalyzed α-pinene SOA in this study. The high-resolution organic fragment families showed an apparent variation pattern with the increase in particle acidity under high-$NO_x$ conditions, as discussed in sections 3.3 and 3.4 of the manuscript. We have added a statement regarding Figure S2 in the revised manuscript:

"The effect of particle acidity on the chemical composition of α-pinene SOA in high- and low-$NO_x$ experiments is examined from the distribution of organic fragments in the high-resolution organic aerosol mass spectra **(see Figure S2 in the Supplement)**." (Lines 302–304)

[Figure]

Figure S2. High-resolution mass spectra of α-pinene SOA under (a–d) high- and (e) low-NO$_x$ conditions. The mass spectra were averaged on the irradiation times of 1–5 h and 2–12 h under high- and low-NO conditions, respectively. The mass spectra of α-pinene SOA under low-NO$_x$ conditions for acidic particles, which are not presented here, resemble that of ammonium sulfate particles in (e).

*Minor Comments:*

*1.) Page 2, Line 35: Need citation to literature here for this sentence.*

The sentence referred to: "The effect of aerosol acidity on SOA formation is one of the scientific questions currently under open debate."

Response: This sentence was followed by detailed explanation regarding "the scientific questions currently under open debate" in the rest of this paragraph. Therefore, rather than adding a citation here, we have revised the original sentence slightly as follows to make it clearer:

"The effect of aerosol acidity on SOA formation is one of the scientific questions currently under open debate, **as described below**." (Lines 36–37)

*2.) Page 2, Line 38: Do the authors mean the use of "e.g." instead of "i.e." when citing these prior studies on acid-catalyzed chemistry?*

Response: Yes, the "i.e." has been corrected to "e.g.".

*3.) Page 2, Line 39: The authors may want to cite Lin et al. (2012, ES&T) from the Surratt group here.*

Response: We have added Lin et al. (2012) as one of the references:

"The presence of acidic aerosol particles has been reported to enhance the reactive uptake of gas phase organic species and increase SOA yields due to acid-catalyzed reactions (e.g., Garland et al., 2006; Jang et al., 2004; Liggio and Li, 2006; **Lin et al., 2012;** Northcross and Jang, 2007; Surratt et al., 2010; Xu et al., 2015a)." (Lines 39–41)

"Lin, Y.-H., Zhang, Z., Docherty, K. S., Zhang, H., Budisulistiorini, S. H., Rubitschun, C. L., Shaw, S. L., Knipping, E. M., Edgerton, E. S., and Kleindienst, T. E.: Isoprene epoxydiols as precursors to secondary organic aerosol formation: acid-catalyzed reactive uptake studies with authentic compounds, Environ. Sci. Technol., 46, 250–258, doi:10.1021/es202554c, 2012." (Lines 39–41)

*4.) Page 2, Lines 41-42: The authors mean to say the following?*

*"Furthermore, the enhanced formation of SOA and organosulfates has been reported from the acid-catalyzed reactive uptake of epoxide compounds in ambient aerosols that are acidic enough to promote this multiphase chemistry."*

The sentence referred to: "Furthermore, the enhanced formation of SOA, organic sulfates, and epoxide compounds has been reported in ambient environments with an abundance of acidic aerosol particles (Hawkins et al., 2010; Lin et al., 2012; Rengarajan et al., 2011; Zhang et al., 2012; Zhou et al., 2012)…"

Response: This statement proposed by the reviewer is more appropriate except the acid-catalyzed reactive uptake is not only limited to epoxide compounds. Therefore, we have revised the original sentence as follows:

"Furthermore, the enhanced formation of SOA **and organic sulfates has been reported from the acid-catalyzed reactive uptake of VOC oxidation products in ambient aerosols that are acidic enough to promote this multiphase chemistry** (Hawkins et al., 2010; Lin et al., 2012; Rengarajan et al., 2011; Zhang et al., 2012; Zhou et al., 2012)…" (Lines 45–48)

*The studies you cite are fine, but you may want to report recent kinetics studies showing that this is feasible, such as work by Matthew Elrod's group (Oberlin college) as well as Gaston et al. (2014, ES&T) and Riedel et al. (2015, ES&T Lett).*

Response: We have added the following statement and the relevant literatures in the revised manuscript:

"In contrast, recent kinetics studies have demonstrated that particle acidity strongly affects the reactive uptake of isoprene epoxydiols (Gaston et al., 2014; Riedel et al., 2015)." (Lines 44–45)

"Gaston, C. J., Riedel, T. P., Zhang, Z., Gold, A., Surratt, J. D., and Thornton, J. A.: Reactive uptake of an isoprene-derived epoxydiol to submicron aerosol particles, Environ. Sci. Technol., 48, 11178–11186, doi:10.1021/es5034266, 2014." (Lines 485–486)

"Riedel, T. P., Lin, Y., Budisulistiorini, S. H., Gaston, C. J., Thornton, J. A., Zhang, Z., Vizuete, W., Gold, A., and Surratt, J. D.: Heterogeneous reactions of isoprene-derived epoxides: reaction probabilities and molar secondary organic aerosol yield estimates, Environ. Sci. Technol. Lett., 2, 38–42, doi:10.1021/ez500406f, 2015." (Lines 618–620)

*5.) Page 12, Line 368: Should "Noted" be changed to "It should be noted...."*

The sentence referred to: "Noted that a small amount of $C_xH_yN_p^+$ and $C_xH_yO_zN_p^+$ fragments were also observed under low-$NO_x$ conditions, where NO was not added (Figure 6b)."

Response: Yes, we agree with this correction. The original sentence has been revised to:

[revised manuscript text omitted]

**S1  High-NO$_x$ regime assessment**

The Master Chemical Mechanism (MCM v3.3.1, http://mcm.leeds.ac.uk/MCMv3.3.1/home.htt) was incorporated into a box model to assess the NO$_x$ regime for the gas-phase reactions of α-pinene photooxidation under high-NO$_x$ conditions. The box model was constrained with the initial experimental conditions including temperature, pressure, and the concentrations of α-pinene, NO, water vapor, and H$_2$O$_2$ for the individual chamber experiments in this study. The photooxidation reaction of α-pinene was simulated for 6 hours with the box model. The output of the box model was the time series of the concentrations of α-pinene, NO, O$_3$, HO$_2$, and organic peroxy radicals (RO$_2$) (molecule cm$^{-3}$) from each time step with a 1-min resolution. The fraction of RO$_2$ radicals reacted with NO compared to the total reacted RO$_2$ radicals (with NO, HO$_2$, and RO$_2$) was calculated by

$$\frac{k_{NO}[NO]}{k_{NO}[NO] + k_{HO_2}[HO_2] + k_{RO_2}[RO_2]}$$

where $k_{NO}$, $k_{HO2}$, and $k_{RO2}$ are the reaction rates of RO$_2$ + NO, RO$_2$ + HO$_2$ and RO$_2$ + RO$_2$, respectively and [NO], [HO$_2$], and [RO$_2$] are the concentration of NO, HO$_2$, and RO$_2$, respectively. The results from the box model are presented in Figure S1. At the start of the simulations, more than 99% of the RO$_2$ radicals were reacting with NO; while by the end of the experiments (after 6 hours), at least 62% of the RO$_2$ radicals continued to react with NO (Figure S1a). The time series for α-pinene, NO, and O$_3$ from the measurements were reasonably well captured by the box model (Figure S1b, c, and d).

[Figure]

**Figure S1. (a) Fraction of RO$_2$ reacted with NO compared to the total reacted RO$_2$ radicals for high-NO$_x$ experiments with ammonium sulfate and acidic seed particles. The measured and the modeled time series of the concentrations of (b) α-pinene, (c) NO, and (d) O$_3$ for the high-NO$_x$ experiment with ammonium sulfate particles (NH$_4$/SO$_4$ = 2). The variations in time for each species in all experiments with acidic particles under high-NO$_x$ conditions are similar to (b), (c), and (d).**

[Figure]

**Figure S2. High-resolution mass spectra of α-pinene SOA under (a–d) high- and (e) low-NO$_x$ conditions. The mass spectra were averaged on the irradiation times of 1–5 h and 2–12 h under high- and low-NO conditions, respectively. The mass spectra of α-pinene SOA under low-NO$_x$ conditions for acidic particles, which are not presented here, resemble that of ammonium sulfate particles in (e).**